# Communication-Efficient Heterogeneous Federated Learning with Generalized Heavy-Ball Momentum

**Riccardo Zaccone**[*]
*Politecnico di Torino*
*riccardo.zaccone@polito.it*

**Sai Praneeth Karimireddy**
*USC Viterbi School of Engineering*
*karimire@usc.edu*

**Carlo Masone**
*Politecnico di Torino*
*carlo.masone@polito.it*

**Marco Ciccone**
*Vector Institute*
*marco.ciccone@vectorinstitute.ai*

**Reviewed on OpenReview:** *https://openreview.net/forum?id=LNoFjcLywb*

## Abstract

Federated Learning (FL) has emerged as the state-of-the-art approach for learning from decentralized data in privacy-constrained scenarios. However, system and statistical challenges hinder its real-world applicability, requiring efficient learning from edge devices and robustness to data heterogeneity. Despite significant research efforts, existing approaches often degrade severely due to the joint effect of heterogeneity and partial client participation. In particular, while momentum appears as a promising approach for overcoming statistical heterogeneity, in current approaches its update is biased towards the most recently sampled clients. As we show in this work, this is the reason why it fails to outperform FEDAVG, preventing its effective use in real-world large-scale scenarios. In this work, we propose a novel *Generalized Heavy-Ball Momentum* (GHBM) and theoretically prove it enables convergence under unbounded data heterogeneity in *cyclic partial participation*, thereby advancing the understanding of momentum's effectiveness in FL. We then introduce adaptive and communication-efficient variants of GHBM that match the communication complexity of FEDAVG in settings where clients can be *stateful*. Extensive experiments on vision and language tasks confirm our theoretical findings, demonstrating that GHBM substantially improves state-of-the-art performance under random uniform client sampling, particularly in large-scale settings with high data heterogeneity and low client participation[1].

## 1 Introduction

Federated Learning (FL) (McMahan et al., 2017) is a paradigm to learn from decentralized data in which a central server orchestrates an iterative two-step training process that involves 1) local training, potentially on a large number of clients, each with its own private data, and 2) the aggregation of these updated local models on the server into a single, shared global model. This process is repeated over several communication rounds. While the inherent privacy-preserving nature of FL makes it well-suited for decentralized applications with

---

[*]Corresponding author
[1]Code is available at `https://github.com/RickZack/GHBM`

restricted data sharing, it also introduces significant challenges. Since local data reflects unique characteristics of individual clients, limiting the optimization to a client's personal data can lead to issues caused by *statistical heterogeneity*. This becomes particularly problematic when multiple optimization steps are performed before model synchronization, causing clients to *drift* from the ideal global updates (Karimireddy et al., 2020). Indeed, heterogeneity has been shown to hinder the convergence of FEDAVG (Hsu et al., 2019), increasing the number of communication rounds needed to achieve a target model quality (Reddi et al., 2021) and negatively impacting final performance.

Several studies have proposed solutions to mitigate the effects of heterogeneity. For instance, SCAF-FOLD (Karimireddy et al., 2020) relies on additional control variables to correct the local client's updates, while FEDDYN (Acar et al., 2021) uses ADMM to align the global and local client solutions. Albeit theoretically grounded, experimentally these methods are not sufficiently robust to handle extreme heterogeneity, low client participation, or large-scale problems, exhibiting slow convergence and instabilities (Varno et al., 2022).

Momentum-based FL methods show promise in addressing these challenges. By accumulating past update directions, momentum can help clients overcome the inconsistencies of local objectives introduced by heterogeneous data. Several works explored incorporating momentum in FL, either at the server (Hsu et al., 2019) or at client-level to correct local updates Ozfatura et al. (2021); Xu et al. (2021). Notably, MIME (Karimireddy et al., 2021) has been proposed as a framework to make clients mimic the updates of a centralized model trained on i.i.d. data by leveraging extra server statistics at the client side. While the theoretical advantages of momentum in FL have been demonstrated under *full participation* Cheng et al. (2024), it has been shown, both theoretically and experimentally, that its effectiveness is limited when client participation varies across training rounds. Indeed, the only momentum-based FL method that operates under *partial participation* and does not rely on assumptions on bounded gradient heterogeneity, SCAFFOLD-M (Cheng et al., 2024), still relies on variance reduction - similarly to SCAFFOLD - to contrast heterogeneity. As a result, it inherits both the limitations of variance reduction in deep learning (Defazio & Bottou, 2019) and the drawbacks of SCAFFOLD in FL, as highlighted by Reddi et al. (2021). In practice, as our work shows, existing momentum-based FL methods exhibit significant limitations in settings with low participation, high heterogeneity, and real-world large-scale problems. Moreover, current approaches often incur increased communication costs due to the additional information exchanged to correct local updates (Karimireddy et al., 2020; 2021; Xu et al., 2021; Ozfatura et al., 2021). This can be a significant drawback in communication-constrained environments, further hindering the practical adoption of FL in real-world applications and highlighting the critical need for more robust, effective, and communication-efficient FL algorithms. In this work, we provide a theoretical justification for the ineffectiveness of classical momentum in FL demonstrating that due to the interplay of data heterogeneity and partial participation, the momentum term is updated with a biased estimate of the global gradient, reducing its effectiveness in correcting client drift. To address these challenges, we propose a novel *Generalized Heavy-Ball* (GHBM) formulation, which computes momentum as a decayed average of the past $\tau$ momentum terms. This design reduces bias toward the most recently selected clients, enabling convergence under arbitrary heterogeneity, not only in full participation but also in *cyclic partial participation*. We then propose FEDHBM, an adaptive and communication-efficient instantiation of GHBM, and experimentally demonstrate its significantly improved performance over state-of-the-art methods.

**Contributions.** We summarize our main results below.

- We present a novel formulation of momentum called *Generalized Heavy-Ball* (GHBM) momentum, which extends the classical heavy-ball (Polyak, 1964), and propose variants that are robust to heterogeneity and communication-efficient by design.
- We establish the theoretical convergence rate of GHBM for non-convex functions, extending the previous result of Cheng et al. (2024) of classical momentum, showing that GHBM converges under arbitrary heterogeneity even (and most notably) in *cyclic partial participation*.
- We empirically show that existing FL algorithms suffer severe limitations in extreme non-iid scenarios and real-world settings. In contrast, GHBM is extremely robust and achieves higher model quality with significantly faster convergence speeds than other client-drift correction methods.

## 2 Related works

**The Problem of Statistical Heterogeneity.** The detrimental effects of non-iid data in FL were first observed by (Zhao et al., 2018), who proposed mitigating performance loss by broadcasting a small portion of public data to reduce the divergence between clients' distributions. Alternatively, (Li & Wang, 2019) uses server-side public data for knowledge distillation. Both approaches rely on the strong assumption of readily available and suitable data. Recognizing weight divergence as a source of performance loss, FEDPROX (Li et al., 2020) adds a regularization term to penalize divergence from the global model. Nevertheless, this was proved ineffective in addressing data heterogeneity Caldarola et al. (2022). Other works (Kopparapu & Lin, 2020; Zaccone et al., 2022; Zeng et al., 2022; Caldarola et al., 2021) explored grouping clients based on their data distribution to mitigate the challenges of aggregating divergent models.

**Stochastic Variance Reduction in FL.** Stochastic variance reduction techniques have been applied in FL (Chen et al., 2021; Li et al., 2019) with SCAFFOLD Karimireddy et al. (2020) providing for the first time convergence guarantees for arbitrarily heterogeneous data. The authors also shed light on the *client-drift* of local optimization, which results in slow and unstable convergence. SCAFFOLD uses control variates to estimate the direction of the server model and clients' models and to correct the local update. This approach requires double the communication to exchange the control variates, and it is not robust enough to handle large-scale scenarios akin to cross-device FL (Reddi et al., 2021; Karimireddy et al., 2021). Similarly, SCAFFOLD-M (Cheng et al., 2024) integrates classical momentum into SCAFFOLD to attain a slightly better convergence rate and maintain robustness to unbounded heterogeneity in partial participation. However, it still relies on variance reduction to tackle heterogeneity, inheriting and the same limitations of SCAFFOLD, as the ineffectiveness of variance reduction in deep learning (Defazio & Bottou, 2019).

**ADMM and Adaptivity.** Other methods are based on the Alternating Direction Method of Multipliers (Chen et al., 2022; Gong et al., 2022; Wang et al., 2022). In particular, FEDDYN(Acar et al., 2021) dynamically modifies the loss function such that the model parameters converge to stationary points of the global empirical loss. Although technically it enjoys the same convergence properties of SCAFFOLD without suffering from its increased communication cost, in practical cases it has displayed problems in dealing with pathological non-iid settings (Varno et al., 2022). Other works explored the use of adaptivity to speed up the convergence of FedAvg and reduce the communication overhead (Xie et al., 2019; Reddi et al., 2021).

**Use of Momentum as Local Correction.** As a first attempt, Hsu et al. (2019) adopted momentum at server-side to reduce the impact of heterogeneity. With a similar idea, Kim et al. (2024) use the Nesterov Accelerated Gradient (NAG) to broadcast a lookahead global model and adds a proximal local penalty similar to FEDPROX (additional details in Appendix A.1). However, server-side momentum has been proven of limited effectiveness under high heterogeneity, because the drift happens at the client level. This motivated later approaches that apply server momentum at each local step (Ozfatura et al., 2021; Xu et al., 2021), and the more general approach by Karimireddy et al. (2021) to adapt any centralized optimizer to cross-device FL. It employs a combination of control variates and server optimizer state (*e.g.* momentum) at each client step, which lead to increased communication bandwidth and frequency. A recent similar approach (Das et al., 2022) employs quantized updates, still requiring significantly more computation client-side. Rather differently from previous works, we propose a novel formulation of momentum specifically designed to take incorporate the descent information of clients selected at past $\tau$ rounds, which generalizes the classical heavy-ball (Polyak, 1964). Most notably, we prove that our GHBM algorithm converges under arbitrary heterogeneity in cyclic partial participation - the first momentum method achieving this result without relying on other mechanisms like variance reduction.

**Lowering Communication Requirements in FL.** Researchers have studied methods to reduce the memory needed for exchanging gradients in the distributed setting, for example by quantization (Alistarh et al., 2017) or by compression (Mishchenko et al., 2019; Koloskova et al., 2020). In the context of FL, such ideas have been developed to meet the communication and scalability constraints (Reisizadeh et al., 2020), and to take into account heterogeneity (Sattler et al., 2020). Our work focuses on a novel formulation of momentum that takes into account the joint effects of heterogeneity and partial participation, and that has a heavy-ball structure allowing efficient use of the information already being sent in vanilla FEDAVG, so additional techniques to compress that information remain orthogonal to our approach.

## 3 Method

### 3.1 Setup

In FL a server and a set $\mathcal{S}$ of clients collaboratively solve a learning problem, with $|\mathcal{S}| = K \in \mathbb{N}^+$. At each round $t \in [T]$, a fraction of $C \in (0, 1]$ clients from $\mathcal{S}$ is selected to participate to the learning process: we denote this portion as $\mathcal{S}^t \subseteq \mathcal{S}$. Each client $i \in \mathcal{S}^t$ receives the server model $\theta_i^{t,0} \equiv \theta^{t-1}$, and performs $J$ local optimization steps, using stochastic gradients $\tilde{g}_i^{t,j}$ evaluated on local parameters $\theta_i^{t,j-1}$ and a batch $d_{i,j}$, sampled from its local dataset $\mathcal{D}_i$. During local training, $\theta_i^{t,j}$ is the model of client $i$ at round $t$ after the $j$-th optimization step, while $\theta_i^t \equiv \theta^{t,J_i}$ is the model sent back to the server. The server then aggregates the client updates $\tilde{g}_i^t := (\theta^{t-1} - \theta_i^t)$, building *pseudo-gradients* $\tilde{g}^t$ that are used to update the model (Reddi et al., 2021).

In this work we formalize the learning objective as a finite-sum optimization problem, where each function is the local clients' loss function with only access to that client's stochastic samples:

$$\arg\min_{\theta \in \mathbb{R}^d} \left[ f(\theta) := \frac{1}{|\mathcal{S}|} \sum_{i \in \mathcal{S}} \left( f_i(\theta) := \mathbb{E}_{d_i \sim \mathcal{D}_i}[f_i(\theta; d_i)] \right) \right] \tag{1}$$

The analysis we provide in Sec. 4.3 is based on the above formalization of the learning problem, which is commonly used to model *cross-silo* FL settings, hence our theoretical results apply to that kind of scenarios. In this context, we prove that GHBM converges under unbounded heterogeneity relying solely on momentum, expanding the understanding of its effectiveness compared to other methods that rely on *variance reduction* or ADMM to achieve this result (Karimireddy et al., 2020; Cheng et al., 2024; Acar et al., 2021). On the other hand, it has been proved that it is not possible to guarantee convergence under arbitrary heterogeneity in the *"stochastic"* or *"streaming"* context which is commonly used for modeling *cross-device* FL (see the lower bound in Theorem 3.4 of Patel et al. (2022)), so considering it in our formal analysis would be of limited usefulness. Hence, we focus the theoretical analysis on the former case. Nevertheless, we also provide large-scale experimental validation on settings that adhere to the characteristics of *cross-device* FL to demonstrate that GHBM is suitable for such real-world scenarios (see Sec. 3.4).

### 3.2 Addressing Client Drift with Momentum

One of the core propositions of federated optimization is to take advantage of local clients' work, by running multiple optimization steps on local parameters before synchronization. This has been proven effective for speeding up convergence when local datasets are i.i.d. with respect to a global distribution (Stich, 2019; Lin et al., 2020; McMahan et al., 2017), and is particularly important for improving communication efficiency, which is the bottleneck when learning in decentralized settings. However, the statistical heterogeneity of clients' local datasets causes local models to *drift* from the ideal trajectory of server parameters. One way of addressing such drift is to use momentum during local optimization, based on the idea that a moving average of past server pseudo-gradients can correct local optimization towards the solution of the global problem. At each round, FL methods based on momentum typically use the gradients of the selected clients, whether computed at local (Xu et al., 2021; Ozfatura et al., 2021) or global (Karimireddy et al., 2021) parameters, to update the momentum term server-side.

**Partial Participation and Biased Momentum.** We claim that existing momentum-based methods overlook a critical aspect of federated learning: *partial client participation*. Indeed, when only a portion of clients participate in the training rounds, the server pseudo-gradient used to update the momentum estimate can be biased towards the previously selected clients, hampering its corrective benefit to local optimization. This effect is particularly pronounced in settings with high data heterogeneity and low client participation (common in cross-device FL), where, as our experiments demonstrate, conventional momentum fails to correct the drift and improve over vanilla FedAvg.

**Main Contribution.** To address the challenges posed by partial participation, we propose a novel momentum-based approach that explicitly accounts for client sampling. Our key idea is to update the momentum term using a pseudo-gradient that approximates the true global gradient over all clients, including those not participating in the current round. By integrating the descent directions from past rounds into local updates, our method effectively mitigates the bias introduced by partial participation, resulting in a

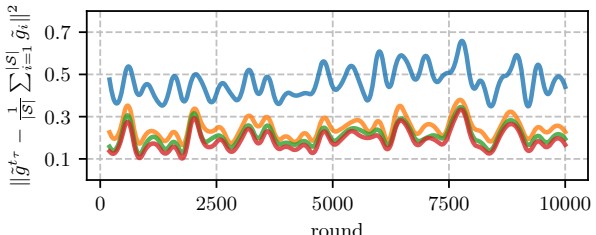 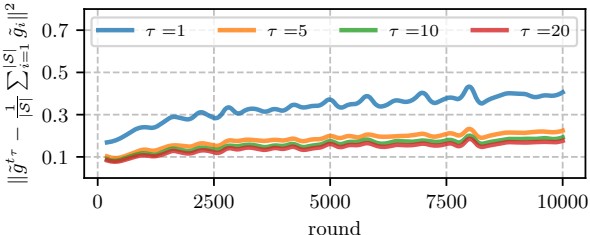

Figure 1: **Reusing old gradients is beneficial, despite the introduced lag.** The plot shows the empirical measure of the deviation between (i) the average of the last $\tau$ server pseudo-gradient (at different parameters) and (ii) the server-pseudo gradient calculated over all the clients (at the same parameters), varying $\tau$, on CIFAR-100 with RESNET-20, in non-iid ($\alpha = 0$, left) and iid ($\alpha = 10.000$, right) settings.

more accurate and robust momentum estimate. Notably, our momentum formulation retains a heavy-ball structure similar to classical momentum, enabling it to be used in FL without requiring to send additional data from server to clients, thus maintaining the same communication complexity as FedAvg.

### 3.3 Generalized Heavy-Ball Momentum (GHBM)

In this section, we introduce our novel formulation for momentum, which we call *Generalized Heavy-Ball Momentum* (GHBM). First, we recall that classical momentum consists of a moving average of past gradients, and it is commonly expressed as in Eq. (2), which can be equivalently expressed in a version commonly referred to as *heavy-ball momentum* in Eq. (3) (see Lemma B.1):

**Heavy-Ball Momentum (HBM)**

$$\tilde{m}^t \leftarrow \beta \tilde{m}^{t-1} + \tilde{g}^t(\theta^{t-1}; \mathcal{D}^t) \quad (2)$$
$$\theta^t \leftarrow \theta^{t-1} - \eta \tilde{m}^t$$

$$\tilde{m}^t \leftarrow (\theta^{t-1} - \theta^{t-2}) \quad (3)$$
$$\theta^t \leftarrow \theta^{t-1} - \eta \tilde{g}^t(\theta^{t-1}; \mathcal{D}^t) + \beta \tilde{m}^t$$

Let us notice that, when applied to FL optimization, the gradient referred to above as $\tilde{g}^t$ is built from updates of clients $i \in \mathcal{S}^t$ (and so on dataset $\mathcal{D}^t := \cup_{i \in \mathcal{S}^t} \mathcal{D}_i$), which are usually a small portion of all the clients participating in the training. Consequently, at each round the momentum is updated using a direction biased towards the distribution of clients selected in that round. Indeed, the prerequisites for this update to reflect the objectives of the other clients are (i) iidness of local datasets or (ii) high client participation. Both conditions are rarely met in practice, and lead to ineffectiveness of existing momentum-based FL methods in realistic scenarios. Our objective is to update the momentum term at each round with a reliable estimate of the gradient w.r.t. the global data distribution of all clients. In practice, the desired update rule for momentum would use the average gradient of all clients selected in the last $\tau$ rounds at current parameters $\theta^{t-1}$, as in Eq. (4).

**Desired Momentum Update**

$$\tilde{m}^t \leftarrow \beta \tilde{m}^{t-1} + \frac{1}{\tau} \sum_{k=t-\tau+1}^{t} \tilde{g}^k(\theta^{t-1}; \mathcal{D}^k) \quad (4)$$

**Practical Momentum Update**

$$\tilde{m}^t \leftarrow \beta \tilde{m}^{t-1} + \frac{1}{\tau} \sum_{k=t-\tau+1}^{t} \tilde{g}^k(\theta^{k-1}; \mathcal{D}^k) \quad (5)$$

While Eq. (4) cannot be implemented in partial participation because clients selected in rounds $k \in [t-\tau+1, t]$ do not have access to model parameters $\theta^{t-1}$, it is possible to reuse old gradients calculated at parameters $\theta^{k-1}$ as their approximation, as shown in Eq. (5). This introduces a *lag* due to using outdated gradients. However, as we show Fig. 1, the benefits of reducing heterogeneity greatly compensate for this lag, as increasing $\tau$ leads to a reduction in the deviation from the gradient calculated over all the clients.

With this idea in mind, our proposed formulation consists of calculating the momentum term as the decayed average of past $\tau$ momentum terms, instead of explicitly using the server pseudo-gradients at the last $\tau$ rounds, as shown in Eq. (6). This formulation is close to the update rule sketched in Eq. (5) and has the additional advantage of enjoying a heavy-ball form similar to Eq. (3) (see Lemma B.2), which will be useful for deriving communication-efficient FL algorithms. In practice, the difference w.r.t. Eq. (3) consists in considering a delta $\tau > 1$:

**Generalized Heavy-Ball Momentum (GHBM)**

$$\tilde{m}_\tau^t \leftarrow \frac{1}{\tau} \sum_{k=1}^{\tau} \beta \tilde{m}_\tau^{t-k} + \tilde{g}^t(\theta^{t-1}; \mathcal{D}^t) \qquad (6)$$
$$\theta^t \leftarrow \theta^{t-1} - \eta \tilde{m}_\tau^t$$

$$\tilde{m}_\tau^t \leftarrow \frac{1}{\tau} \left( \theta^{t-1} - \theta^{t-\tau-1} \right) \qquad (7)$$
$$\theta^t \leftarrow \theta^{t-1} - \eta \tilde{g}^t(\theta^{t-1}; \mathcal{D}^t) + \beta \tilde{m}_\tau^t$$

As it is trivial to notice, GHBM with $\tau = 1$ recovers the classical momentum, hence it can be considered as a generalized formulation. The GHBM term is then embedded into local updates using the heavy-ball form shown in Eq. (7), leading to the following update rule:

$$\textsc{Client step:} \qquad \theta_i^{t,j} \leftarrow \theta_i^{t,j-1} - \eta_l \tilde{g}_i^{t,j}(\theta_i^{t,j-1}; d_i^{t,j}) + \underbrace{\frac{\beta}{\tau J} \left( \theta^{t-1} - \theta^{t-\tau-1} \right)}_{\tau - \mathrm{GHBM}} \qquad (8)$$

**Discussion on $\tau$.** The $\tau$ hyperparameter in GHBM plays a crucial role, since it controls the number of server pseudo-gradients to average when estimating the update to the momentum term. Intuitively, when considering only the effect on heterogeneity reduction, the optimal value would be the one that provides the average over all clients. Under proper assumptions on client sampling (see Sec. 4.1), this optimal value is $\tau = 1/C$, which is the inverse of the client participation rate. As we demonstrate, this property is the key factor that allows GHBM to converge under arbitrary heterogeneity, achieving the same convergence rate in *cyclic partial participation* as methods based on classical momentum attain in *full participation* (see Sec. 4.3). However, because GHBM reuses old gradients, it introduces a *lag* that grows with $\tau$. Therefore, the optimal choice of $\tau$ comes with an inevitable trade-off between the heterogeneity reduction effect and other sources of error, which we discuss in Sec. 4.2.

### 3.4 Communication Complexity of GHBM and Efficient Variants

---
**Algorithm 1:** GHBM, LocalGHBM and FedAvg

---
**Require:** initial model $\theta^0$, $K$ clients, $C$ participation ratio, $T$ number of total round, $\eta$ and $\eta_l$ learning rates, $\tau \in \mathbb{N}^+$.
1: **for** $t = 1$ to $T$ **do**
2: $\quad \mathcal{S}^t \leftarrow$ subset of clients $\sim \mathcal{U}(\mathcal{S}, \max(1, K \cdot C))$
3: $\quad$ Send $\theta^{t-1}$, $\theta^{t-\tau-1}$ to all clients $i \in \mathcal{S}^t$
4: $\quad$ **for** $i \in \mathcal{S}^t$ **in parallel do**
5: $\qquad \theta_i^{t,0} \leftarrow \theta^{t-1}$
6: $\qquad$ Retrieve $\theta^{t-\tau_i-1}$ from local storage
7: $\qquad \tilde{m}_\tau^t \leftarrow \frac{1}{\tau J}(\theta^{t-1} - \theta^{t-\tau-1})$
8: $\qquad \tilde{m}_{\tau_i}^t \leftarrow \frac{1}{\tau_i J}(\theta^{t-1} - \theta^{t-\tau_i-1})$ **if** $\theta^{t-\tau_i-1}$ is set **else 0**
9: $\qquad$ **for** $j = 1$ to $J$ **do**
10: $\qquad\quad$ sample a mini-batch $d_{i,j}$ from $\mathcal{D}_i$
11: $\qquad\quad \theta_i^{t,j} \leftarrow \theta_i^{t,j-1} - \eta_l \tilde{g}_i^{t,j} + \beta \tilde{m}_\tau^t + \beta \tilde{m}_{\tau_i}^t$
12: $\qquad$ **end for**
13: $\qquad$ Save model $\theta^{t-1}$ into local storage
14: $\quad$ **end for**
15: $\quad \tilde{g}^t \leftarrow \frac{1}{|\mathcal{S}^t|} \sum_{i \in \mathcal{S}^t} \left( \theta^{t-1} - \theta_i^{t,J} \right)$
16: $\quad \theta^t \leftarrow \theta^{t-1} - \eta \tilde{g}^t$
17: **end for**

---

As it is possible to notice from Algorithm 1, GHBM requires the server to additionally send the past model $\theta^{t-\tau-1}$, which is used to calculate the momentum term in Eq. (8). Alternatively, the server could send the momentum term $\tilde{m}_\tau^t$: in both cases, this introduces a communication overhead of $1.5\times$ w.r.t. FedAvg, as momentum is usually applied to all model parameters. However, this overhead can be avoided by leveraging the observation that the choice of $\tau = 1/C$ is expected to be optimal. Indeed, it is sufficient to notice that, if clients participate cyclically, *i.e.*, the period between each subsequent sampling is equal for all clients, and the frequency at which each client is selected for training is exactly $1/C$. Notice that this is still true on average under uniform client sampling, *i.e.*, calling $\tau_i$ the sampling period for client $i$, $\mathbb{E}[\tau_i] = \tau = 1/C$.

Leveraging those observations and exploiting the fact that GHBM has an equivalent heavy-ball form, the additional requirement on communication can be traded for a requirement on persistent storage at the clients, allowing them to keep the model received by the server across rounds, as shown in Algorithm 1. In this algorithm, which we call **LocalGHBM**, $\tau_i$ is adaptive and determined stochastically by client participation. The space complexity is constant in the size of model parameters for the clients and the communication complexity is the same as FedAvg. We empirically found that performance can be further improved by considering $\theta_{i,j}^t$ instead of $\theta^{t-1}$ and $\theta_i^{t-\tau_i}$ instead of $\theta^{t-\tau_i-1}$ when calculating $\tilde{m}_{\tau_i}^t$. This final communication-efficient update rule is named **FedHBM**.

Table 1: **Comparison of convergence rates of FL algorithms.** GHBM improves the state-of-art by attaining, in *cyclic partial participation*, the same rate of classical momentum in *full participation*. Remind that $L$ is the smoothness constant of objective functions, $\Delta = f(\theta^0) - \min_\theta f(\theta)$ is the initialization gap, $\sigma^2$ is the clients' gradient variance, $|\mathcal{S}|$ is the number of clients, $C$ is the participation ratio, $J$ is the number of local steps per round, and $T$ is the number of communication rounds. $\zeta := \sup_\theta \|\nabla f(\theta)\|$ and $G$ are uniform bounds of gradient norm and dissimilarity.

| Algorithm | Convergence Rate $\frac{1}{T}\sum_{t=1}^{T} \mathbb{E}\left[\|\nabla f(\theta^t)\|^2\right] \lesssim$ | Additional Assumptions | Partial participation? |
|---|---|---|---|
| FEDAVG (Yang et al., 2021) | $\left(\frac{L\Delta\sigma^2}{|\mathcal{S}|JT}\right)^{1/2} + \frac{L\Delta}{T}$ | Bounded hetero.[1] | ✗ |
| (Yang et al., 2021) | $\left(\frac{L\Delta J\sigma^2}{|\mathcal{S}|CT}\right)^{1/2} + \frac{L\Delta}{T}$ | Bounded hetero.[1] | ✓ |
| FEDCM (Xu et al., 2021) | $\left(\frac{L\Delta(\sigma^2+|\mathcal{S}|CJ\zeta^2)}{|\mathcal{S}|CJT}\right)^{1/2} + \left(\frac{L\Delta(\sigma/\sqrt{J}+\sqrt{|\mathcal{S}|C}(\zeta+G))}{\sqrt{|\mathcal{S}|CT}}\right)^{2/3}$ | Bounded grad. Bounded hetero. | ✓ |
| (Cheng et al., 2024) | $\left(\frac{L\Delta\sigma^2}{|\mathcal{S}|JT}\right)^{1/2} + \frac{L\Delta}{T}$ | – | ✗ |
| SCAFFOLD-M (Cheng et al., 2024) | $\left(\frac{L\Delta\sigma^2}{|\mathcal{S}|CJT}\right)^{1/2} + \frac{L\Delta}{T}\left(1 + \frac{|\mathcal{S}|^{2/3}}{|\mathcal{S}|C}\right)$ | – | ✓ |
| **GHBM (Thm. 4.11)** | $\left(\frac{L\Delta\sigma^2}{|\mathcal{S}|JT}\right)^{1/2} + \frac{L\Delta}{T}$ | Cyclic participation | ✓ |

[1] The local learning rate vanishes to zero when gradient dissimilarity is unbounded, *i.e.*, $G \to \infty$.

**Applicability of GHBM-based Algorithms in FL Scenarios.** Although based on the same principle, our algorithms are suitable for different scenarios. Similarly to algorithms proposed for cross-device FL (Karimireddy et al., 2021), GHBM uses *stateless* clients, with the main $\tau$ hyperparameter controlled by the server. This ensures that clients always apply a momentum term consistent with the GHBM update rule, differently from algorithms that require clients participating in multiple rounds to adhere to their formulation, such as SCAFFOLD and FEDDYN. This is particularly important when the number of clients is large and a small portion of them participates in each round, and it is why, in our large-scale setting, these methods fail to converge. These design choices make our algorithm in practice suitable for cross-device FL, where it offers significant advantages, as experimentally validated in Sec. 5.3. On the other hand, FEDHBM and LOCALGHBM take advantage of the fact that clients participate multiple times in the training process to remove the need to send the momentum term from the server, recovering the same communication complexity of FEDAVG. As a result, clients in these methods are *stateful* - requiring to maintain variables across rounds (Kairouz et al., 2021) - and are therefore best suited for scenarios akin to *cross-silo* FL.

## 4 Theoretical Discussion

In this section, we establish the theoretical foundations of our algorithms. Our analysis reveals that: (i) the momentum update rule implemented by GHBM in Eq. (5) approximates an update with global gradient, with $\tau$ controlling the trade-off between heterogeneity reduction and the *lag* due to using old gradients; (ii) thanks to this algorithmic design choice, GHBM converges under arbitrary heterogeneity even in (cyclic) partial participation. The proofs are deferred to Appendix B.

### 4.1 Assumptions

To prove our results we rely on notions of stochastic gradient with bounded variance (4.1) and the smoothness of the clients' objective functions (4.2), which are common in deep learning. Additionally, to facilitate comparisons with other algorithms that require it, we introduce the Bounded Gradient Dissimilarity (BGD) (Assumption 4.3). This assumption, commonly used in FL literature, provides an upper bound on the dissimilarity of clients' objectives. While our main result in Thm. 4.11 does not require this assumption, we use it to demonstrate the heterogeneity reduction effect of GHBM, and to show that, under the proper choice of $\tau$, BGD is not necessary. Finally, we introduce the additional assumption that clients participate following a cyclic pattern (Assumption 4.4). Notably, this assumption is only required for obtaining our convergence rate and serves as a technical detail needed to deterministically quantify the contributions of the clients to the GHBM momentum term (see Fig. 6 in the Appendix for an illustration of cyclic participation).

**Assumption 4.1** (Unbiasedness and bounded variance of stochastic gradient)**.**

$$\mathbb{E}_{d_i \sim \mathcal{D}_i} [\tilde{g}_i(\theta; d_i)] = g_i(\theta; \mathcal{D}_i)$$

$$\mathbb{E}_{d_i \sim \mathcal{D}_i} \left[ \|\tilde{g}_i(\theta; d_i) - g_i(\theta; \mathcal{D}_i)\|^2 \right] \leq \sigma^2$$

**Assumption 4.2** (Smoothness of client's objectives)**.** Let it be a constant $L > 0$, then for any $i$, $\theta_1$, $\theta_2$ the following holds:

$$\|g_i(\theta_1) - g_i(\theta_2)\|^2 \leq L^2 \|\theta_1 - \theta_2\|^2$$

**Assumption 4.3** (Bounded Gradient Dissimilarity)**.** There exist a constant $G \geq 0$ such that, $\forall i, \theta$:

$$\frac{1}{|\mathcal{S}|} \sum_{i=1}^{|\mathcal{S}|} \|g_i(\theta) - g(\theta)\|^2 \leq G^2$$

**Assumption 4.4** (Cyclic Participation)**.** Let $\mathcal{S}^t$ be the set of clients sampled at any round $t$. A sampling strategy is *"cyclic"* with period $p = 1/C$ if:

$$\mathcal{S}^t = \mathcal{S}^{t-p} \quad \forall\, t > p \quad \wedge \quad \mathcal{S}^k \cap \mathcal{S}^t = \varnothing \quad \forall\, k \in (t-p, t)$$

*Remark* 4.5. **Our main result (Thm. 4.11) does not require the BGD assumption**: indeed we show that, under a proper choice of $\tau$, the effect of heterogeneity is completely removed from the convergence rate.

*Remark* 4.6. While Thm. 4.11 relies on Assumption 4.4, **cyclic participation is not enforced in the experiments**, where we select clients randomly and uniformly, ensuring fair comparison with algorithms that do not need this assumption in their analysis. For a more comprehensive discussion on the role of the cyclic participation assumption in our work, we refer the reader to Sec. 4.3.

### 4.2 Overcoming Bounded Gradient Dissimilarity in Partial Participation

In this section, we explain the core elements used in our theory to guarantee convergence under arbitrary heterogeneity for GHBM.

**Bounding the Participation-induced Heterogeneity.** Let us recall the main idea behind GHBM: because of partial participation, at each round classical momentum is updated using a direction biased towards the distribution of clients selected in that round. As a result, recalling that GHBM recovers classical momentum when $\tau = 1$, we begin by bounding the effect of heterogeneity induced by partial client participation on the momentum estimate as a function of $\tau$. To this end, let us provisionally adopt Assumption 4.3 and assume we perform federated optimization with a single full gradient step in partial participation and consider the momentum update in Eq. (4). In this setup, the following lemma holds:

**Lemma 4.7** (Deviation of $\tau$-averaged gradient from true gradient)**.** *Define $\mathcal{S}_\tau^t := \cup_{k=0}^{\tau-1} \mathcal{S}^{t-k}$ as the set of clients selected in the last $\tau$ rounds, and $g^{t_\tau} := 1/|\mathcal{S}_\tau^t| \sum_{i=1}^{|\mathcal{S}_\tau^t|} g_i^t(\theta^{t-1})$ as the average server pseudo-gradient. The approximation of a gradient over the last $\tau$ rounds $g^{t_\tau}$ w.r.t. the true gradient is quantified by the following:*

$$\mathbb{E}\left[ \|g^{t_\tau} - \nabla f(\theta^{t-1})\|^2 \right] \leq 8\mathbb{E}\left[ \left( \frac{|\mathcal{S}| - |\mathcal{S}_\tau^t|}{|\mathcal{S}|} \right)^2 \right] \left( G^2 + \|\nabla f(\theta^{t-1})\|^2 \right)$$

Lemma 4.7 shows that, as $\tau$ increases, the effect of heterogeneity reduces quadratically as the difference between the $|\mathcal{S}^t|$ and $|\mathcal{S}_\tau^t|$ approaches to zero. The deviation is exactly zero when $\mathcal{S}_\tau^t = \mathcal{S}$, *i.e.* the set of clients selected in the last $\tau$ rounds includes all the clients. While under uniform sampling it is unlikely to realize this condition because of the non-zero probability of sampling the same clients over consecutive rounds, under cyclic participation it is possible to make the above error exactly equal to zero [2].

**Corollary 4.8.** *Consider Lemma 4.7 and further assume that, at each round of FL training, clients are sampled according to a rule satisfying Assumption 4.4. Then, for any $\tau \in \left(0, \frac{1}{C}\right]$:*

$$\mathbb{E}\left[ \|g^{t_\tau} - \nabla f(\theta^{t-1})\|^2 \right] \leq 8\left(1 - \tau C\right)^2 \left( G^2 + \|\nabla f(\theta^{t-1})\|^2 \right)$$

*Remark* 4.9. Under Assumption 4.4 and $\tau = 1/C$, **the BGD assumption (4.3) is not necessary**, as the two terms in the left-hand side (LHS) of the above inequality are the same by definition.

---

[2]An alternative approach could keep track of gradients of each client and then compute $g^{t_\tau}$ such that it includes the latest gradients of all clients. In that case, cyclic participation is not necessary, but calculating the needed $\tau$ is an instance of the *Batched Coupons Collector* problem (Stadje, 1990; Ferrante & Frigo, 2012; Ferrante & Saltalamacchia, 2014), for which a closed form solution is unknown. That approach would be unrealistic to implement so, motivated by the strong empirical success of GHBM, in our analysis we prefer adopting an additional assumption, and providing guarantees under cyclic client participation

**Bounding the Overall Error in Momentum Update.** In the previous paragraph, we established the role of $\tau$ in GHBM for counteracting heterogeneity and derived its optimal value w.r.t. partial client participation. However, our analysis assumed that all clients selected in the last $\tau$ rounds compute a full gradient on the same server parameters. As discussed in Sec. 3.3, a more realistic update rule for momentum would reuse past gradients as in Eq. (5), computed at local parameters. This is because clients selected in rounds $k \in [t - \tau + 1, t)$ do not have access to model parameters $\theta^{t-1}$. As a result, increasing $\tau$ introduces additional sources of error to the momentum term, which we quantify in the following lemma.

**Lemma 4.10** (Bounded Error of Momentum Update). *Consider the update rule in Eq. (5), and call* $\tilde{g}^{t_\tau} = \frac{1}{\tau} \sum_{k=t-\tau+1}^{t} \frac{1}{|\mathcal{S}^k|J} \sum_{i=1}^{|\mathcal{S}^k|} \sum_{j=1}^{J} \tilde{g}_i^{k,j}(\theta_i^{k,j-1})$ *the server stochastic average pseudo-gradient over the last* $\tau$ *global steps and the average server pseudo-gradient at current parameters as* $g^{t_\tau} := 1/|\mathcal{S}_\tau^t| \sum_{i=1}^{|\mathcal{S}_\tau^t|} g_i^t(\theta^{t-1})$. *Let also define the client drift* $\mathcal{U}_t := \frac{1}{|\mathcal{S}|J} \sum_{j=1}^{J} \sum_{i=1}^{|\mathcal{S}|} \mathbb{E}\|\theta_i^{t,j} - \theta^{t-1}\|^2$ *and the error of server update* $\mathcal{E}_t := \mathbb{E}\|\nabla f(\theta^{t-1}) - \tilde{m}_\tau^{t+1}\|^2$. *Under Assumptions 4.1, 4.2 and 4.4, it holds that:*

$$
\mathbb{E}\left[\left\|\tilde{g}^{t_\tau} - g^{t_\tau}\right\|^2\right] \leq 3\left( \underbrace{\frac{\sigma^2}{|\mathcal{S}_\tau^t|J}}_{(a)\ Noise} + \underbrace{\frac{L^2}{\tau} \sum_{k=t-\tau+1}^{t} \mathcal{U}_k}_{(b)\ Client\ drift} + \underbrace{2L^2\eta^2 \sum_{k=t-\tau+1}^{t-1} \left( \mathbb{E}\left[\left\|\nabla f(\theta^{k-1})\right\|^2\right] + \mathcal{E}_k \right)}_{(c)\ Gradient\ lag} \right)
$$

Lemma 4.10 shows that the error affecting the GHBM momentum update rule can be decomposed into three main components: the first term **(a)** is caused by clients taking stochastic gradients on mini-batches of data. The dependency indicates that increasing $\tau$ has a positive effect until the gradients of all clients participate to the estimate (*i.e.* $\mathcal{S}_\tau^t = \mathcal{S}$). The second term **(b)** represents the average client drift over the last $\tau$ rounds, arising from clients performing multiple local steps. The lemma shows this term has a benign dependency, as increasing $\tau$ does not increase the overall error due to this component. The last term **(c)** is the *gradient lag*, which reflects the error introduced by using pseudo-gradients from clients based on old parameters. While this may be the main source of error (since it linearly increases with $\tau$), it depends on $\mathcal{E}_k$, which is the deviation of server update from the true gradient. If momentum succeeds in correcting local optimization (*i.e.* $\mathcal{E}_k$ is small), this term will also be small and not hinder the optimization. We verify experimentally that this is indeed the case: the heterogeneity reduction achieved by increasing $\tau$ outweighs the overall error bounded in Lemma 4.10, as showed in Fig. 1.

### 4.3 Convergence Guarantees

We can now state the convergence result for GHBM for ***non-convex*** functions in (cyclic) partial participation. Comparison with recent related algorithms is provided in Tab. 1.

**Theorem 4.11.** *Under Assumptions 4.1, 4.2 and 4.4, if we take* $\tilde{m}_\tau^0 = 0$, *and* $\beta$, $\eta$ *and* $\eta_l$ *as in Eq. (120), then GHBM with* $\tau = 1/C$ *converges as:*

$$
\frac{1}{T} \sum_{t=1}^{T} \mathbb{E}\left[\left\|\nabla f(\theta^{t-1})\right\|^2\right] \lesssim \frac{L\Delta}{T} + \sqrt{\frac{L\Delta\sigma^2}{|\mathcal{S}|JT}}
$$

*where* $\Delta := f(\theta^0) - \min_\theta f(\theta)$, $\eta_l \leq \mathcal{O}\left(1/\sqrt{\tau}\right)$ *(see Eq. (120)) and* $\lesssim$ *absorbs numeric constants.*

**Discussion.** The rate of GHBM shows two major improvements: (i) it does not rely on the BGD assumption (4.3) and (ii) the dominant term on the right-hand side (RHS) scales with the size of all client population $|\mathcal{S}|$, instead of the clients selected in a single round $|\mathcal{S}|C$, thanks to incorporating old gradients. While under the assumptions of Thm. 4.11 any $\tau = \frac{k}{C}$, $\forall k \in \mathbb{N}^+$ will lead to similar conclusions, considering larger interval increases the error due to using old gradients (see Sec. 4.2), so we would like to choose $\tau$ as the minimum allowing convergence under unbounded heterogeneity. Indeed, a larger $\tau$ imposes a stricter bound on the client learning rate $\eta_l \leq \mathcal{O}\left(1/\sqrt{\tau}\right)$ in Eq. (120). Since Thm. 4.11 also imposes $\tau = 1/C$, the bound on $\eta_l$ is explicitly related to the participation ratio $C$.

**Comparison with FedCM.** The best-known rate for FEDCM in partial participation (Xu et al., 2021) relies both on bounded gradients and bounded gradient dissimilarity and it is asymptotically weaker than ours. For the case of *full participation*, Cheng et al. (2024) proved that FEDCM converges without requiring bounded client dissimilarity. Our results extend theirs in that we prove that GHBM can achieve the same convergence rate even in cyclic partial participation. This follows from the fact that in this setting GHBM update rule approximates the one of classical momentum in full participation. Indeed, to validate this theoretical finding, in Figure 2 we simulate a cyclic participation setting and show the train loss of GHBM across rounds, comparing with FEDCM, both when selecting a subset of clients and when selecting them all. As it is shown, the curve of GHBM with $\tau$ as prescribed by Thm. 4.11 approaches the one of FEDCM in full participation.

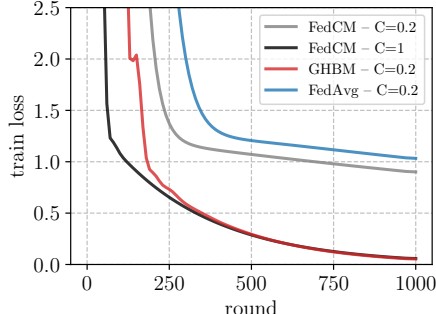

Figure 2: **Comparison between FedCM and GHBM** in *cyclic participation* on a linear regression problem, non-iid setting, with $J = 2$ local steps and $K = 10$ clients. GHBM with $\tau = 1/C$ in *cyclic participation* ($C = 0.2$) performs similarly as FEDCM in *full participation* ($C = 1$).

**Comparison with SCAFFOLD-M.** Recently Cheng et al. (2024) proved that momentum accelerates SCAFFOLD, preserving strong guarantees against heterogeneity in partial participation. However, the resulting SCAFFOLD-M method is still based on variance reduction, *i.e.*, it converges under arbitrary heterogeneity thanks to variance reduction, not because it uses momentum. Our rate additionally requires Assumption 4.4, but is faster and, most importantly, shows that momentum, when modified according to our formulation, can by itself provide similar guarantees even when not all clients participate.

**Advantage of Local Steps and Connections to Incremental Gradient Methods.** Thm. 4.11 does not show an explicit benefit from the local steps, similar to the best-known theory for momentum-based FL methods (Cheng et al., 2024). However, GHBM offers a clear advantage w.r.t. centralized methods for finite-sum optimization applied in FL (where clients represent functions), referred to as *incremental gradient methods*. One algorithm of this family, the Incremental Aggregated Gradient (IAG), removes the effect of functions heterogeneity by approximating a full gradient with an aggregate of past gradients, assuming cyclic participation (Gürbüzbalaban et al., 2015). However, this holds only in standard distributed mini-batch optimization, where $J = 1$. GHBM shares a similar intuition, but applying this logic to the momentum update rather than the gradient estimate is crucial when local steps are involved. Simply extending IAG with local steps would not mitigate client drift-induced heterogeneity as GHBM does. In fact, its convergence rate would be bounded by that of FEDAVG in full participation, whose lower bound is known to be affected by heterogeneity (see Thm. II of Karimireddy et al. (2020)).

**On the Use of Cyclic Participation Assumption.** The use of cyclic participation in the proof of Thm. 4.11 allows precise control over the clients' contributions to the average of the last $\tau$ pseudo-gradients. This ensures that the $\tau$-averaged pseudo-gradient used to update the momentum is unaffected by heterogeneity, which is the important point behind the proof of Thm. 4.11. Under random uniform, due to the non-zero probability of sampling the same client within $\tau$ rounds, this condition is hardly verified. Although one could technically enforce this condition without cyclic sampling — by explicitly tracking each client's pseudo-gradient and computing a uniform average across the most recent one from each client — this would be impractical. Such a design would not be compliant with protocols like Secure Aggregation, widely adopted in real-world FL systems, thus posing a significant practical limitation.

Please note that in our analysis convergence under unbounded heterogeneity is not a simple byproduct of the assumption, but comes explicitly from the algorithmic structure of GHBM (*i.e.* setting $\tau = \frac{k}{C}$, $\forall k \in \mathbb{N}^+$ is **necessary**). The best-known analysis of FEDAVG under cyclic participation is provided by Cho et al. (2023), which proves that in certain situations (*e.g.* clients run GD instead of SGD) there can be an asymptotic advantage in the case we prospect with Assumption 4.4. However, it is important to notice that all the results presented in Cho et al. (2023) rely on forms of bounded heterogeneity, and with this respect, the results presented in this work are novel and advance state of the art.

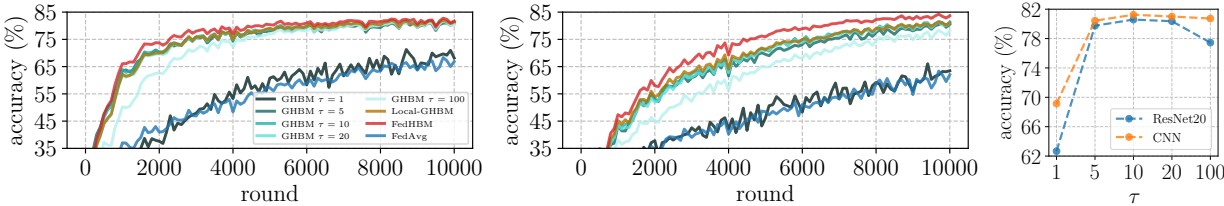

Figure 3: **GHBM effectively counteracts the effects of heterogeneity:** our momentum formulation ($\tau > 1$) is crucial for superior performance, with an optimal value $\tau = 1/C = 10$, as predicted in theory. Results on CIFAR-10 with CNN (left) and RESNET-20 (right), under worst-case heterogeneity.

## 5 Experimental Results

We present evidence both in controlled and real-world scenarios, showing that: (i) the GHBM formulation is pivotal to enable momentum to provide an effective correction even in extreme heterogeneity, (ii) our adaptive LOCALGHBM effectively exploits client participation to enhance communication efficiency and (iii) GHBM is suitable for cross-device scenarios, with stark improvement on large datasets and architectures.

### 5.1 Setup

**Scenarios, Datasets and Models.** For the controlled scenarios, we employ CIFAR-10/100 as computer vision tasks, with RESNET-20 and the same CNN similar to a LeNet-5 commonly used in FL works (Hsu et al., 2020), and SHAKESPEARE dataset as NLP task following (Reddi et al., 2021; Karimireddy et al., 2021). For CIFAR-10/100 we follow the common practice of Hsu et al. (2020), sampling local datasets according to a Dirichlet distribution with concentration parameter $\alpha$, denoting as NON-IID and IID respectively the splits corresponding to $\alpha = 0$ and $\alpha = 10.000$ (additional details in Appendix C.2). For SHAKESPEARE we use instead the predefined splits (Caldas et al., 2019). The datasets are partitioned among $K = 100$ clients, selecting a portion $C = 10\%$ of them at each round. The training round budget $T$ is set to be big enough for all algorithms to reach convergence in the worst-case scenario ($\alpha = 0$), constrained by a time budget for the simulations. Being our proposed algorithm always faster, this ensures fair comparison with competitors.

For simulating real-world scenarios, we adopt the large-scale GLDv2 and INATURALIST datasets as CV tasks, with both a VIT-B\16 (Dosovitskiy et al., 2021) and a MOBILENETV2 (Sandler et al., 2018) pretrained on ImageNet, and STACKOVERFLOW dataset as NLP task, following Reddi et al. (2021); Karimireddy et al. (2021). These settings are particularly challenging, because the learning tasks are complex, the number of client is high (*i.e.* on the order of $10^4$-$10^5$) and the client participation (for convenience directly reported in Tab. 3) is scarce (see details in Tab. 6). As is, those settings are akin to *cross-device* FL.

**Metrics and Experimental protocol.** As metrics, we consider *final model quality*, as the average top-1 accuracy over the last 100 rounds of training (Tabs. 2 and 3), and *communication/computational efficiency*: this is evaluated by measuring the total amount of exchanged bytes (*i.e.* considering both the downlink and uplink communication) and the wall-clock time spent by an algorithm to reach the performance of FEDAVG (Tab. 4). We also provide full convergence curves for a subset of the experiments in Fig. 5. Results are always reported as the average over 5 independent runs, performed on the best-performing hyperparameters extensively searched separately for all competitor algorithms. **All the experiments are conducted under random uniform client sampling**, as it is standard practice. Further details on datasets, splits, models and hyperparameters are in Appendix C.

### 5.2 The Effectiveness of GHBM Compared to Classical Momentum

We provide evidence of the effectiveness of GHBM under worst-case heterogeneity (*i.e.* $\alpha = 0$) by comparing the impact of our generalized heavy-ball momentum formulation to the classical momentum approach, which corresponds to selecting $\tau > 1$ in the update rule in Eq. (8). As shown in Fig. 3, prior momentum-based methods (Xu et al., 2021; Ozfatura et al., 2021) fail to improve upon FEDAVG. In contrast, as $\tau$ increases, GHBM exhibits a significant enhancement in both convergence speed and final model quality. The optimal value of $\tau$ is experimentally determined to be $\tau \approx 1/C = 10$, with larger sub-optimal values only slightly affecting performance (rightmost plot).

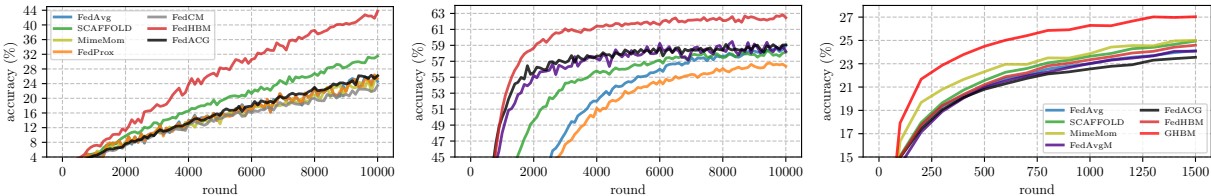

Figure 5: **GHBM largely outperforms state-of-the-art methods:** the plots show the test accuracy (%) over rounds, with RESNET-20 on CIFAR-100, both in NON-IID (left) and IID (middle) settings, and on STACKOVERFLOW (right). GHBM always displays much faster convergence and higher accuracy, even when distributions are IID, confirming robustness w.r.t. heterogeneity and better dependency on stochastic noise.

This experiment demonstrates that, while complete heterogeneity reduction is theoretically proven only under cyclic participation (*i.e.* Thm. 4.11 holds under Assumption 4.4), GHBM empirically achieves strong heterogeneity reduction even with random uniform client sampling. In particular, the theoretical prescription on the optimal value $\tau = {}^1/C$ also holds in this setting. Moreover, our communication-efficient variants always match or surpass the best-tuned GHBM, confirming that their adaptive estimate of each client's momentum positively contributes in a scenario of stochastic client participation (see Sec. 4.2).

### 5.3 Comparison with the State-of-art

**Results in Controlled Scenario.** We compare GHBM with the most common FL methods, and in particular with other momentum-based FL algorithms, including the recently proposed SCAFFOLD-M (Cheng et al., 2024), which which uses both the control variates of SCAFFOLD and the momentum of FEDCM (and consequently incurs in a communication overhead of $2.5\times$ w.r.t. FEDAVG). Our results in Tab. 2 underscore that methods based on classical momentum fail at improving FEDAVG in scenarios with high heterogeneity and partial participation, confirming that in those cases they should not be expected to provide a significant advantage over heterogeneity. The general ineffectiveness of classical momentum also holds for SCAFFOLD-M: as it is possible to notice, its performance is not significantly better than SCAFFOLD's, and this well aligns with the theory, where the guarantees against heterogeneity come from the use of control variates, while momentum only brings acceleration. In that our results align with previous findings in literature suggesting that variance reduction, besides theoretically strong, is often not effective empirically in deep learning (Defazio & Bottou, 2019). Conversely, our algorithms outperform FEDAVG with an impressive margin of $+20.6\%$ and $+14.4\%$ on RESNET-20 and CNN under worst-case heterogeneity, and consistently over less severe conditions (higher values of $\alpha$ in Fig. 4). In particular, as shown in Fig. 5, GHBM improves over competitor methods also in IID scenarios: this relates to our convergence rate improving not only w.r.t. heterogeneity, but also displaying a better dependency on the stochastic noise.

Table 2: **Comparison with state-of-the-art in controlled setting** (acc@10$k$-20$k$ rounds for RESNET-20/CNN). NON-IID ($\alpha = 0$) and IID ($\alpha = 10.000$). Best result in **bold**, second best underlined. ✗ indicates non-convergence.

| METHOD | CIFAR-100 (RESNET-20) | | CIFAR-100 (CNN) | | SHAKESPEARE | |
|---|---|---|---|---|---|---|
| | NON-IID | IID | NON-IID | IID | NON-IID | IID |
| FEDAVG | 24.7 ±1.2 | 58.6 ±0.4 | 38.3 ±0.3 | 49.7 ±0.2 | 47.3 ±0.1 | 47.1 ±0.2 |
| FEDPROX | 24.8 ±1.1 | 58.5 ±0.3 | 40.6 ±0.2 | 49.9 ±0.2 | 47.3 ±0.1 | 47.1 ±0.2 |
| SCAFFOLD | 30.7 ±1.3 | 58.0 ±0.6 | 45.5 ±0.1 | 49.4 ±0.4 | 50.2 ±0.1 | 50.1 ±0.1 |
| FEDDYN | 6.0 ±0.5 | 60.8 ±0.7 | ✗ | 51.9 ±0.2 | 50.7 ±0.2 | 50.8 ±0.2 |
| ADABEST | 8.4 ±2.0 | 55.6 ±0.3 | 35.6 ±0.3 | 49.7 ±0.2 | 47.3 ±0.1 | 47.1 ±0.2 |
| MIME | 26.8 ±2.1 | 59.0 ±0.3 | 45.3 ±0.4 | 50.9 ±0.4 | 48.3 ±0.2 | 48.5 ±0.1 |
| FEDAVGM | 24.8 ±0.7 | 58.7 ±0.9 | 42.1 ±0.3 | 50.7 ±0.2 | 50.0 ±0.0 | 50.4 ±0.1 |
| FEDACG | 25.7 ±0.5 | 58.7 ±0.3 | 43.5 ±0.4 | 51.3 ±0.3 | 50.9 ±0.1 | 51.0 ±0.1 |
| SCAFFOLD-M | 30.9 ±0.7 | 60.1 ±0.5 | 45.7 ±0.2 | 50.1 ±0.3 | 50.8 ±0.0 | 51.0 ±0.1 |
| FEDCM (GHBM $\tau=1$) | 22.2 ±1.0 | 53.1 ±0.2 | 36.0 ±0.3 | 50.2 ±0.5 | 49.2 ±0.1 | 50.4 ±0.1 |
| FEDADC (GHBM $\tau=1$) | 22.4 ±0.1 | 53.2 ±0.2 | 37.9 ±0.3 | 50.2 ±0.4 | 49.2 ±0.1 | 50.4 ±0.1 |
| MIMEMOM | 24.3 ±0.9 | 60.5 ±0.6 | 48.2 ±0.7 | 50.6 ±0.1 | 48.5 ±0.2 | 48.9 ±0.2 |
| MIMELITEMOM | 21.2 ±1.6 | 59.2 ±0.5 | 46.0 ±0.3 | 50.7 ±0.1 | 49.1 ±0.4 | 49.4 ±0.3 |
| **LocalGHBM (ours)** | 38.2 ±1.0 | 62.0 ±0.5 | 50.3 ±0.5 | 51.9 ±0.4 | 51.2 ±0.1 | 51.1 ±0.3 |
| **FedHBM (ours)** | **42.5** ±0.8 | **62.5** ±0.5 | **50.4** ±0.5 | **52.0** ±0.4 | **51.3** ±0.1 | **51.4** ±0.2 |

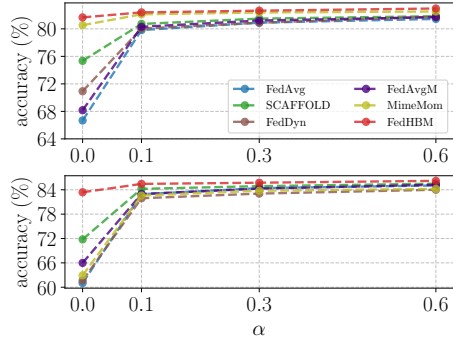

Figure 4: **Final model quality at different values of $\alpha$** (lower $\alpha \to$ higher heterogeneity) on CIFAR-10, with CNN (top) and RESNET-20 (bottom).

**Results in Real-world Large-scale Scenarios.** Extending the experimentation to settings characterized by extremely low client participation, we test both our GHBM with $\tau$ tuned via a grid-search and our adaptive FEDHBM, which exploits client participation to keep the same communication complexity of FEDAVG. As discussed in Secs. 3.3 and 4.2, under such extreme client participation patterns GHBM performs better because the trade-off between heterogeneity reduction and gradient lag is explicitly tuned by the choice of the best performing $\tau$, while FEDHBM will likely adopt a suboptimal value. However, results in Tab. 3 show a stark improvement over the state-of-art for both our algorithms, indicating that the design principle of our momentum formulation is remarkably robust and provides effective improvement even when client participation is very low (*e.g.* $C \le 1\%$).

Table 3: **Test accuracy (%) comparison of best SOTA FL algorithms on large-scale and realistic settings.** GHBM is the best algorithm when client participation is extremely low, while FEDHBM still improves the other competitors by a large margin. ✗ means that the algorithm did not converge.

| METHOD | MOBILENETV2 | | | | VIT-B\16 | | | STACKOVERFLOW |
|---|---|---|---|---|---|---|---|---|
| | GLDv2 | INATURALIST | | | GLDv2 | INATURALIST | | |
| | $C \approx 0.79\%$ | $C \approx 0.1\%$ | $C \approx 0.5\%$ | $C \approx 1\%$ | $C \approx 0.79\%$ | $C \approx 0.1\%$ | $C \approx 0.5\%$ | $C \approx 0.12\%$ |
| FEDAVG | 60.3 ±0.2 | 38.0 ±0.8 | 45.25 ±0.1 | 47.59 ±0.1 | 68.5 ±0.5 | 65.6 ±0.1 | 70.7 ±0.8 | 24.0 ±0.4 |
| SCAFFOLD | 61.0 ±0.1 | ✗ | ✗ | ✗ | 67.5 ±3.3 | ✗ | ✗ | 24.8 ±0.4 |
| FEDAVGM | 61.5 ±0.2 | 41.3 ±0.4 | 46.0 ±0.1 | 48.4 ±0.1 | 70.0 ±0.5 | 66.0 ±0.2 | 71.4 ±0.5 | 24.1 ±0.3 |
| MIMEMOM | ✗ | ✗ | ✗ | ✗ | ✗ | ✗ | ✗ | 24.9 ±0.6 |
| **GHBM - best $\tau$ (ours)** | **65.9** ±0.1 | **41.8** ±0.1 | **48.7** ±0.1 | **50.5** ±0.1 | **74.3** ±0.6 | **68.8** ±0.3 | **73.5** ±0.4 | **27.0** ±0.1 |
| **FedHBM (ours)** | 65.4 ±0.2 | 41.6 ±0.2 | 47.3 ±0.0 | 49.8 ±0.0 | 73.1 ±0.9 | 66.7 ±0.7 | 72.1 ±0.5 | 24.5 ±0.4 |

**Communication Efficiency.** Results in Tab. 4 reveal that our proposed algorithms lead to a dramatic reduction in both communication and computational cost, with an average saving of respectively $+55.9\%$ and $+61.5\%$. In practice, while FEDHBM has the same communication complexity of FEDAVGM and GHBM slightly higher, both our algorithms much show faster convergence and higher final model quality, which ultimately lead to a significant reduction of the total communication and computational cost. In particular, in settings with extremely low client participation (*e.g.* GLDv2 and INATURALIST), GHBM is more suitable for best accuracy, while FEDHBM is the best at lowering the communication cost.

Table 4: **Total communication and computational cost for reaching the final model quality of FedAvg**, across academic and real-world large-scale datasets (details in Appendix C.3). The coloured arrows indicate respectively a reduction ($\downarrow$) and an increase ($\uparrow$) of communication/computational cost.

| METHOD | COMM. OVERHEAD | TOTAL COMMUNICATION COST (BYTES EXCHANGED) | | | | TOTAL COMPUTATIONAL COST (WALL-CLOCK TIME HH:MM) | | | |
|---|---|---|---|---|---|---|---|---|---|
| | | CIFAR-100 ($\alpha=0$) | | GLDv2 | | CIFAR-100 ($\alpha=0$) | | GLDv2 | |
| | | CNN | RESNET-20 | MOBILENETV2 | VIT-B\16 | CNN | RESNET-20 | MOBILENETV2 | VIT-B\16 |
| FEDAVG | 1× | 30.9 GB | 10.3 GB | 89.8 GB | 483.7 GB | 02:05 | 03:36 | 13:51 | 13:56 |
| SCAFFOLD | 2× | 40.8 GB ↑32.0% | 14.2 GB ↑37.8% | 51.2 GB ↓43.0% | 967.4 GB ↑100.0% | 01:23 ↓34.0% | 02:39 ↓26.4% | 08:28 ↓38.9% | 15:15 ↑9.4% |
| FEDAVGM | 1× | 21.0 GB ↓32.0% | 9.1 GB ↓11.6% | 73.6 GB ↓18.0% | 403.1 GB ↓16.7% | 01:25 ↓32.0% | 03:10 ↓12.0% | 11:22 ↓18.0% | 11:37 ↓16.7% |
| MIMEMOM | 3× | 21.5 GB ↓30.4% | 30.9 GB ↑200.0% | 269.4 GB ↑200.0% | 1.417 TB ↑200.0% | 01:27 ↓30.4% | 10:42 ↑197.8% | 41:07 ↑197.8% | 41:30 ↑197.8% |
| **GHBM (ours)** | 1.5× | 8.5 GB ↓72.5% | 7.0 GB ↓32.5% | 48.5 GB ↓46.0% | 314.4 GB ↓35.0% | 00:24 ↓80.8% | 01:37 ↓55.0% | 05:20 ↓61.5% | 06:30 ↓53.3% |
| **FedHBM (ours)** | 1× | **5.2 GB** ↓83.0% | **4.2 GB** ↓59.2% | **29.6 GB** ↓67.0% | **234.4 GB** ↓51.5% | **00:22** ↓82.0% | **01:29** ↓59.0% | 06:23 ↓54.0% | 07:31 ↓46.0% |

# 6 Conclusions

In this work, we propose *Generalized Heavy-Ball Momentum* (GHBM), a novel momentum-based optimization method for Federated Learning (FL) that effectively mitigates the joint effect of statistical heterogeneity and partial participation. We theoretically prove that GHBM converges under arbitrary heterogeneity in *cyclic partial participation*, achieving the same rate classical momentum enjoys in *full participation*. Additionally, we introduce FEDHBM, a communication-efficient variant that retains the benefits of momentum while maintaining the same communication complexity as FEDAVG. Extensive experiments, conducted under standard random uniform client sampling, confirm that GHBM significantly outperforms state-of-the-art FL methods in both convergence speed and final model quality, demonstrating its robustness in large-scale, real-world heterogeneous FL scenarios.

## Acknowledgements

The authors would like to thank Carlo Ciliberto for fruitful initial discussions on the theoretical aspects of GHBM and for his valuable feedback about the presentation of the method.

## Funding

The author(s) declare that financial support was received for the research, authorship, and/or publication of this article. This study was carried out within the project FAIR - Future Artificial Intelligence Research - and received funding from the European Union Next-GenerationEU [PIANO NAZIONALE DI RIPRESA E RESILIENZA (PNRR) – MISSIONE 4 COMPONENTE 2, INVESTIMENTO 1.3 – D.D. 1555 11/10/2022, PE00000013 - CUP: E13C22001800001]. This manuscript reflects only the authors' views and opinions, neither the European Union nor the European Commission can be considered responsible for them. A part of the computational resources for this work was provided by hpc@polito, which is a Project of Academic Computing within the Department of Control and Computer Engineering at the Politecnico di Torino (`http://www.hpc.polito.it`). We acknowledge the CINECA award under the ISCRA initiative for the availability of high-performance computing resources. This work was supported by CINI.

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

# A Additional Discussion

## A.1 Extended Related Works

Recently, similarly based on variance reduction as SCAFFOLD, (Mishchenko et al., 2022) propose SCAFFNEW to achieve accelerated communication complexity in heterogeneous settings through control variates, guaranteeing convergence under arbitrary heterogeneity in full participation. The work by Mishchenko et al. (2024), under the assumption of second-order data heterogeneity, proposes an algorithm which can reduce client drift by estimating the global update direction as well as employing regularization. The proposed algorithm can be seen as a combination of FEDPROX with SCAFFOLD/SCAFFNEW, and similarly relies on additional server control variates to correct the drift, so the underlying principle is still variance reduction. Quite differently, GHBM is based on momentum, properly modified to tackle heterogeneity and partial participation in FL. Similarly to the already discussed MIME (Karimireddy et al., 2021), Karagulyan et al. (2024) propose the SPAM algorithm and leverage momentum as a local correction term to benefit from second-order similarity.

**Comparison with FedACG (Kim et al., 2024).** We provide a comparison with the FedACG algorithm based on: algorithmic design, theoretical guarantees and empirical results. Algorithmically, it has two modifications w.r.t. FEDAVGM: (i) it uses the Nesterov Accelerated Gradient (NAG) to broadcast a lookahead global model and (ii) adds a proximal local penalty similar to FEDPROX w.r.t. this transmitted global model. The method has the same communication complexity as FedAvg, because it does not exchange additional information. Our work proposes instead a novel formulation of momentum, explicitly designed to provide an advantage in heterogeneous FL with partial client participation. We propose both the main algorithm (GHBM), which has *stateless* clients but has $1.5\times$ the communication complexity of FedAvg, and communication efficient versions (*e.g.* FEDHBM), that preserve the communication complexity as FedAvg, at the cost of using local storage. From a theoretical perspective, the convergence rate of FedACG does not prove any advantage w.r.t. heterogeneity, since it still relies on the bounded heterogeneity assumption. GHBM is proven to converge under arbitrary heterogeneity in cyclic partial participation, recovering the same convergence rate that Cheng et al. (2024) proved for FEDCM when in full participation. This is a significant advantage that then reflects in significantly improved performance. From an empirical perspective, simulation results are presented in Fig. 5. While it is faster than FedAvgM, it still falls short behind our algorithms in heterogeneous scenarios. This is a consequence of the same issue we showed in Sec. 3.3 for classical momentum.

Cyclic participation with period p=3 for any round k s.t. k mod p = 0

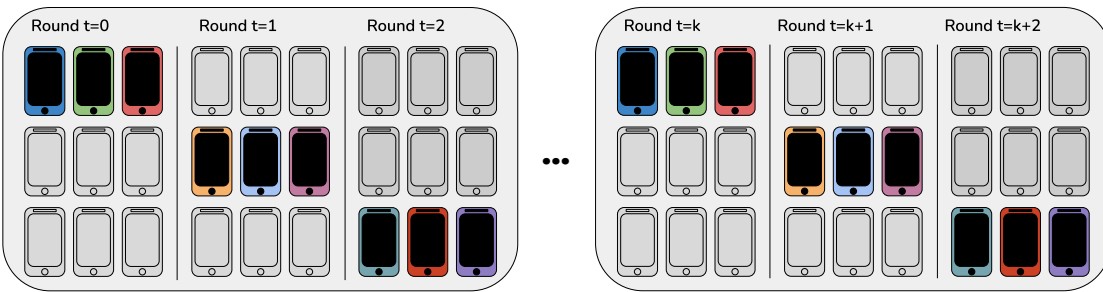

Figure 6: **Illustration of cyclic client participation with a total of $K = 9$ clients.** Thm. 4.11 holds under the assumption of cyclic participation, which simply states that there is any fixed order (so client shuffling methods like Shuffle-Once are compliant with the assumption) in which clients appear across rounds in the training, *i.e.* each client is sampled every $p = \frac{1}{C}$ rounds. In the above image, $K \cdot C = 3$ clients are selected for training, *i.e.* each client is selected exactly once every $p = 3$ rounds.

## A.2 Notes on Failure Cases of SOTA Algorithms

In this paper, we evaluated our approach using the large-scale FL datasets proposed by (Hsu et al., 2020). Notably, several recent state-of-the-art FL algorithms failed to converge on these datasets. For SCAFFOLD this result aligns with prior works (Reddi et al., 2021; Karimireddy et al., 2021), since it is unsuitable for cross-device FL with thousands of devices. Indeed, the client control variates can become stale, and

may consequently degrade the performance. For MimeMom (Karimireddy et al., 2021), despite extensive hyperparameter tuning using the authors' original code, we were unable to achieve convergence. This finding is surprising since the approach has been proposed to tackle cross-device FL. To our knowledge, this is the first work to report these failure cases, likely due to the lack of prior evaluations on such challenging datasets. We believe these findings underscore the need for further investigation into the factors contributing to algorithm performance in large-scale, heterogeneous FL settings.

## B  Proofs

### Algorithms

To handle the proof, we analyze a simpler version of our algorithm, in which we use the update rule in Eq. (5) instead of the one described in Eq. (6). The resulting Algorithm 3 we analyze is reported along the plain GHBM (Algorithm 2) we used in the experiments. Both algorithms enjoy the same underlying idea: use the gradients of a larger portion of the clients to estimate the momentum term.

---

**Algorithm 2:** GHBM (PRACTICAL VERSION)

---

**Require:** initial model $\theta^0$, $K$ clients, $C$ participation ratio, $T$ number of total round, $\eta$ and $\eta_l$ learning rates, $\tau \in \mathbb{N}^+$.

1: **for** $t = 1$ to $T$ **do**
2:     $\mathcal{S}^t \leftarrow$ subset of clients $\sim \mathcal{U}(\mathcal{S}, \max(1, K \cdot C))$
3:     **for** $i \in \mathcal{S}^t$ **in parallel do**
4:         $\theta_i^{t,0} \leftarrow \theta^{t-1}$
5:         **for** $j = 1$ to $J$ **do**
6:             sample a mini-batch $d_{i,j}$ from $\mathcal{D}_i$
7:             $u_i^{t,j} \leftarrow \nabla f_i(\theta_i^{t,j-1}, d_{i,j}) + \beta \tilde{m}_\tau^t$
8:             $\theta_i^{t,j} \leftarrow \theta_i^{t,j-1} - \eta_l u_i^{t,j}$
9:         **end for**
10:    **end for**
11:    $u^t \leftarrow \frac{1}{|\mathcal{S}^t|} \sum_{i \in \mathcal{S}^t} \left( \theta^{t-1} - \theta_i^{t,J} \right)$
12:    $\theta^t \leftarrow \theta^{t-1} - \eta u^t$
13:    $\tilde{m}_\tau^{t+1} \leftarrow \frac{1}{\tau J} \left( \theta^{t-\tau} - \theta^t \right)$
14: **end for**

---

**Algorithm 3:** GHBM (THEORY VERSION)

---

**Require:** initial model $\theta^0$, $K$ clients, $C$ participation ratio, $T$ number of total round, $\eta$ and $\eta_l$ learning rates, $\tau \in \mathbb{N}^+$.

1: **for** $t = 1$ to $T$ **do**
2:     $\mathcal{S}^t \leftarrow$ subset of clients $\sim \mathcal{U}(\mathcal{S}, \max(1, K \cdot C))$
3:     **for** $i \in \mathcal{S}^t$ **in parallel do**
4:         $\theta_i^{t,0} \leftarrow \theta^{t-1}$
5:         **for** $j = 1$ to $J$ **do**
6:             sample a mini-batch $d_{i,j}$ from $\mathcal{D}_i$
7:             $u_i^{t,j} \leftarrow \beta \nabla f_i(\theta_i^{t,j-1}, d_{i,j}) + (1 - \beta) \tilde{m}_\tau^t$
8:             $\theta_i^{t,j} \leftarrow \theta_i^{t,j-1} - \eta_l u_i^{t,j}$
9:         **end for**
10:    **end for**
11:    $u^t \leftarrow \frac{1}{\eta_l |\mathcal{S}^t| J} \sum_{i \in \mathcal{S}^t} \left( \theta^{t-1} - \theta_i^{t,J} \right)$
12:    $\bar{\theta}^t \leftarrow \theta^{t-1} - u^t + (1 - \beta) \tilde{m}_\tau^t$
13:    $\tilde{m}_\tau^{t+1} \leftarrow (1 - \beta) \tilde{m}_\tau^t + \frac{1}{\tau} \left( \bar{\theta}^{t-\tau} - \bar{\theta}^t \right)$
14:    $\theta^t \leftarrow \theta^{t-1} - \eta \tilde{m}_\tau^{t+1}$
15: **end for**

---

In the following, we list the differences between the two:

1. Explicit use of $\tau$-averaged gradients when updating the momentum term (line 13). This can be implemented by keeping server-side an auxiliary sequence of models $\bar{\theta}^t$, in which the momentum added client side is subtracted server-side (line 12), such that taking the difference of two models gives the sum of pseudo-grads.

2. Use of convex sum in local updates (line 7). This is done to align with the formulation of momentum methods in Cheng et al. (2024), and more in general with the formulation of momentum commonly

analyzed in literature. There is no theoretical difference between the two versions, as they only differ by a constant scaling (Liu et al., 2020).

3. Use of gradients averaged over local steps (line 11). This is done to align with the analysis of Cheng et al. (2024); Xu et al. (2021), and it is equivalent to coupling server and client learning rates (*i.e.* setting $\eta = \gamma J \eta_l$ in Algorithm 3, where $\gamma$ is the server learning rate we would use in Algorithm 2).

The two algorithms have similar performances, which are reported in Fig. 7

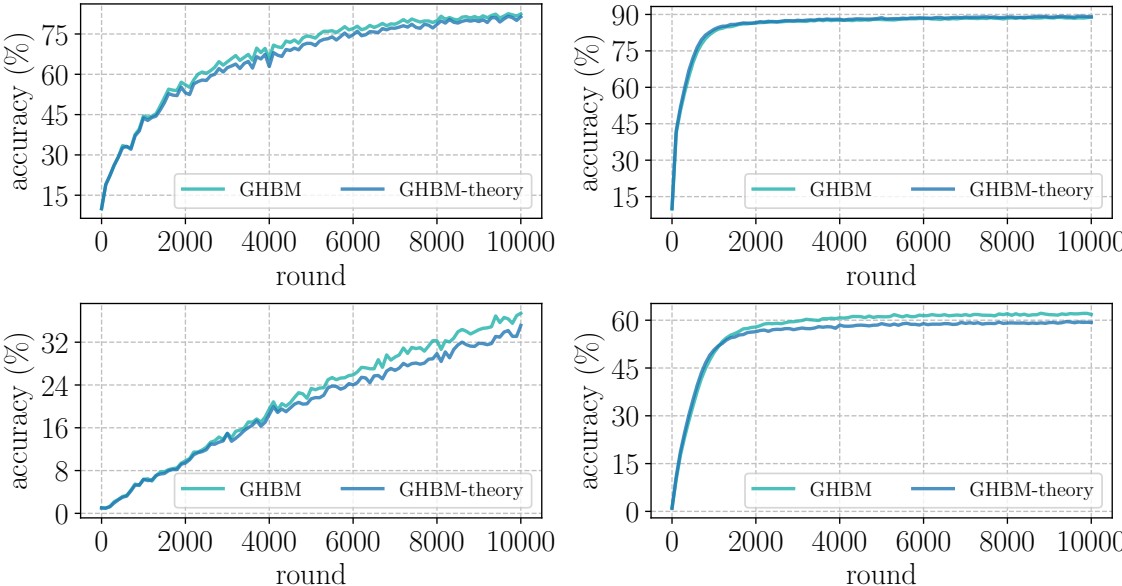

Figure 7: Comparing the GHBM implementation analyzed in theory (Algorithm 3) with the one proposed in the main paper (Algorithm 2). The plots show the convergence rate on CIFAR-10 (top) and CIFAR-100 (bottom), in NON-IID (left) and IID (right) scenarios with RESNET-20 architecture.

**Preliminaries**

Our convergence proof for GHBM is based on the recent work of Cheng et al. (2024), which offers new proof techniques for momentum-based FL algorithms. Throughout the proofs we use the following auxiliary variables to facilitate the presentation:

$$\mathcal{U}_t := \frac{1}{|\mathcal{S}|J} \sum_{j=1}^{J} \sum_{i=1}^{|\mathcal{S}|} \mathbb{E}\left[\left\|\theta_i^{t,j} - \theta^{t-1}\right\|^2\right] \tag{9}$$

$$\mathcal{E}_t := \mathbb{E}\left[\left\|\nabla f(\theta^{t-1}) - \tilde{m}_\tau^{t+1}\right\|^2\right] \tag{10}$$

$$\zeta_i^{t,j} := \mathbb{E}\left[\theta_i^{t,j+1} - \theta_i^{t,j}\right] \tag{11}$$

$$\Xi_t := \frac{1}{|\mathcal{S}|} \sum_{i=1}^{|\mathcal{S}|} \mathbb{E}\left[\left\|\zeta_i^{t,0}\right\|^2\right]$$

$$\Lambda_t := \mathbb{E}\left[\left\|\left(\frac{1}{\tau} \sum_{k=t-\tau+1}^{t} \frac{1}{|\mathcal{S}^k|J} \sum_{i=1}^{|\mathcal{S}^k|} \sum_{j=1}^{J} \tilde{g}_i^{k,j}(\theta_i^{k,j-1})\right) - g^{t_\tau}\right\|^2\right] \tag{12}$$

$$\gamma_t := \mathbb{E}\left[\left\|g^{t_\tau} - \nabla f(\theta^{t-1})\right\|^2\right] \tag{13}$$

Additionally, here we report the *bounded gradient heterogeneity* assumption. It is used to quantify the heterogeneity reduction effect of GHBM varying its $\tau$ hyperparameter. Notice that our main claim does not depend on this assumption, as for the optimal value of $\tau = 1/C$ the assumption is not needed (see Lemma 4.7).

### B.1 Momentum Expressions

In this section we report the derivation of the momentum expressions in Eq. (3) and (7) from the main paper.

**Lemma B.1** (Heavy-Ball Formulation of Classical Momentum). *Let us consider the following classical formulation of momentum:*

$$\tilde{m}^t = \beta \tilde{m}^{t-1} + \tilde{g}^t(\theta^{t-1}) \tag{14}$$

$$\theta^t = \theta^{t-1} - \eta \tilde{m}^t \tag{15}$$

*The same update rule can be equivalently expressed with the following, known as heavy-ball formulation:*

$$\theta^t = \theta^{t-1} + \beta(\theta^{t-1} - \theta^{t-2}) - \eta \tilde{g}(\theta^{t-1}) \tag{16}$$

*Proof.* First derive the expression of $\tilde{m}^t$ from Eq. (15), both for time $t$ and $t-1$:

$$\tilde{m}^t = \frac{\left(\theta^{t-1} - \theta^t\right)}{\eta}$$

$$\tilde{m}^{t-1} = \frac{\left(\theta^{t-2} - \theta^{t-1}\right)}{\eta}$$

Now plug these expressions into Eq. (14) to obtain (16):

$$\frac{\left(\theta^{t-1} - \theta^t\right)}{\eta} = \beta \frac{\left(\theta^{t-2} - \theta^{t-1}\right)}{\eta} + \tilde{g}^t(\theta^{t-1})$$

$$\left(\theta^t - \theta^{t-1}\right) = \beta \left(\theta^{t-1} - \theta^{t-2}\right) - \eta \tilde{g}^t(\theta^{t-1})$$

$$\theta^t = \theta^{t-1} + \beta \left(\theta^{t-1} - \theta^{t-2}\right) - \eta \tilde{g}^t(\theta^{t-1})$$

$\square$

**Lemma B.2** (Heavy-Ball formulation of generalized momentum). *Let us consider the following generalized formulation of momentum:*

$$\tilde{m}^t_\tau = \frac{1}{\tau} \sum_{k=1}^{\tau} \beta \tilde{m}^{t-k}_\tau + \tilde{g}^t(\theta^{t-1}) \tag{17}$$

$$\theta^t = \theta^{t-1} - \eta \tilde{m}^t_\tau \tag{18}$$

*The same update rule can be equivalently expressed in an heavy ball form, which we call as Generalized Heavy-Ball momentum (GHBM):*

$$\theta^t = \theta^{t-1} + \frac{\beta}{\tau}(\theta^{t-1} - \theta^{t-\tau-1}) - \eta \tilde{g}(\theta^{t-1}) \tag{19}$$

*Proof.* First derive the expression of $\tilde{m}^t_\tau$ from Eq. (18), both for time $t$ and $t-1$:

$$\tilde{m}^t_\tau = \frac{\left(\theta^{t-1} - \theta^t\right)}{\eta}$$

$$\tilde{m}^{t-1}_\tau = \frac{\left(\theta^{t-2} - \theta^{t-1}\right)}{\eta}$$

Now plug these expressions into Eq. (17):

$$\frac{\left(\theta^{t-1} - \theta^{t}\right)}{\eta} = \frac{\beta}{\tau} \sum_{k=1}^{\tau} \frac{\left(\theta^{t-k-1} - \theta^{t-k}\right)}{\eta} + \tilde{g}^{t}(\theta^{t-1})$$

$$\left(\theta^{t} - \theta^{t-1}\right) = \frac{\beta}{\tau} \sum_{k=1}^{\tau} \left(\theta^{t-k} - \theta^{t-k-1}\right) - \eta\tilde{g}^{t}(\theta^{t-1})$$

$$\theta^{t} = \theta^{t-1} + \frac{\beta}{\tau} \sum_{k=1}^{\tau} \left(\theta^{t-k} - \theta^{t-k-1}\right) - \eta\tilde{g}^{t}(\theta^{t-1})$$

$$\theta^{t} = \theta^{t-1} + \frac{\beta}{\tau}(\theta^{t-1} - \theta^{t-\tau-1}) - \eta\tilde{g}^{t}(\theta^{t-1})$$

Where the last equality (19) comes from telescoping the summation on the rhs. $\qquad\square$

## B.2 Technical Lemmas

Now we cover some technical lemmas which are useful for computations later on. These are known results that are reported here for the convenience of the reader.

**Lemma B.3** (relaxed triangle inequality). *Let $\{\boldsymbol{v}_1, \dots, \boldsymbol{v}_n\}$ be $n$ vectors in $\mathbb{R}^d$. Then, the following is true:*

$$\left\| \sum_{i=1}^{n} \boldsymbol{v}_i \right\|^2 \le n \sum_{i=1}^{n} \|\boldsymbol{v}_i\|^2$$

*Proof.* By Jensen's inequality, given a convex function $\phi$, a series of $n$ vectors $\{\boldsymbol{v}_1, \dots, \boldsymbol{v}_n\}$ and a series of non-negative coefficients $\lambda_i$ with $\sum_{i=1}^{n} \lambda_i = 1$, it results that

$$\phi\left( \sum_{i=1}^{n} \lambda_i \boldsymbol{v}_i \right) \le \sum_{i=1}^{n} \lambda_i \phi\left(\boldsymbol{v}_i\right)$$

Since the function $\boldsymbol{v} \to \|\boldsymbol{v}\|^2$ is convex, we can use this inequality with coefficients $\lambda_1 = \dots = \lambda_n = 1/n$, with $\sum_{i=1}^{n} \lambda_i = 1$, and obtain that

$$\left\| \frac{1}{n} \sum_{i=1}^{n} \boldsymbol{v}_i \right\|^2 = \frac{1}{n^2} \left\| \sum_{i=1}^{n} \boldsymbol{v}_i \right\|^2 \le \frac{1}{n} \sum_{i=1}^{n} \|\boldsymbol{v}_i\|^2$$

$\qquad\square$

## B.3 Proofs of Main Lemmas

In this section we provide the proofs of the main theoretical results presented in the main paper.

**Proof of Lemma 4.7** (Deviation of $\tau$-averaged gradient from true gradient)

Let define $\mathcal{S}_d := \mathcal{S} - \mathcal{S}_\tau^t$ and $\mathcal{S}_i := \mathcal{S} \cap \mathcal{S}_\tau^t$. Let us note that when all clients participate, *i.e.* $\mathcal{S}_d = \emptyset$, the claim is trivially true. For $\mathcal{S}_d \ne \emptyset$, we can expand the terms at the left-hand side using their definitions as follows:

$$\gamma_t = \mathbb{E}\left[\left\|\frac{1}{|\mathcal{S}_\tau^t|}\sum_{i=1}^{|\mathcal{S}_\tau^t|} g_i^t - \frac{1}{|\mathcal{S}|}\sum_{i=1}^{|\mathcal{S}|} g_i^t\right\|^2\right] \tag{20}$$

$$= \mathbb{E}\left[\left\|\sum_{i\in\mathcal{S}_i}\left(\frac{1}{|\mathcal{S}_\tau^t|} - \frac{1}{|\mathcal{S}|}\right) g_i^t - \sum_{k\in\mathcal{S}_d}\frac{1}{|\mathcal{S}|} g_k^t\right\|^2\right] \tag{21}$$

$$\overset{\text{lemma B.3}}{\leq} 2\left(\underbrace{\mathbb{E}\left[\left\|\sum_{i\in\mathcal{S}_i}\left(\frac{1}{|\mathcal{S}_\tau^t|} - \frac{1}{|\mathcal{S}|}\right) g_i^t\right\|^2\right]}_{\mathcal{T}_3} + \underbrace{\mathbb{E}\left[\left\|\sum_{k\in\mathcal{S}_d}\frac{1}{|\mathcal{S}|} g_k^t\right\|^2\right]}_{\mathcal{T}_4}\right) \tag{22}$$

Let us consider first $\mathcal{T}_3$. We have:

$$\mathcal{T}_3 = \mathbb{E}\left[\left\|\sum_{i\in\mathcal{S}_i}\left(\frac{1}{|\mathcal{S}_\tau^t|} - \frac{1}{|\mathcal{S}|}\right) g_i^t\right\|^2\right] = \mathbb{E}\left[\left(\frac{1}{|\mathcal{S}_\tau^t|} - \frac{1}{|\mathcal{S}|}\right)^2\left\|\sum_{i\in\mathcal{S}_i} g_i^t\right\|^2\right] \tag{23}$$

$$\overset{\text{lemma B.3}}{\leq} \mathbb{E}\left[\left(\frac{1}{|\mathcal{S}_\tau^t|} - \frac{1}{|\mathcal{S}|}\right)^2 |\mathcal{S}_i|\sum_{i\in\mathcal{S}_i}\|g_i^t\|^2\right] \tag{24}$$

$$= \mathbb{E}\left[\left(\frac{1}{|\mathcal{S}_\tau^t|} - \frac{1}{|\mathcal{S}|}\right)^2 |\mathcal{S}_i|\sum_{i\in\mathcal{S}_i}\|g_i^t - \nabla f(\theta^{t-1}) + \nabla f(\theta^{t-1})\|^2\right] \tag{25}$$

$$\overset{\text{lemma B.3}}{\leq} 2\mathbb{E}\left[\left(\frac{1}{|\mathcal{S}_\tau^t|} - \frac{1}{|\mathcal{S}|}\right)^2 |\mathcal{S}_i|\sum_{i\in\mathcal{S}_i}\left(\|g_i^t - \nabla f(\theta^{t-1})\|^2 + \|\nabla f(\theta^{t-1})\|^2\right)\right] \tag{26}$$

$$\overset{\text{assumption 4.3}}{\leq} 2\mathbb{E}\left[\left(\frac{1}{|\mathcal{S}_\tau^t|} - \frac{1}{|\mathcal{S}|}\right)^2 |\mathcal{S}_i|\left(|\mathcal{S}_i|G^2 + \sum_{i\in\mathcal{S}_i}\|\nabla f(\theta^{t-1})\|^2\right)\right] \tag{27}$$

Since the term $\nabla f(\theta^{t-1})$ does not depend on the index $i$, we get

$$2\mathbb{E}\left[\left(\frac{1}{|\mathcal{S}_\tau^t|} - \frac{1}{|\mathcal{S}|}\right)^2 |\mathcal{S}_i|\left(|\mathcal{S}_i|G^2 + \sum_{i\in\mathcal{S}_i}\|\nabla f(\theta^{t-1})\|^2\right)\right] \tag{28}$$

$$= 2\mathbb{E}\left[\left(\frac{1}{|\mathcal{S}_\tau^t|} - \frac{1}{|\mathcal{S}|}\right)^2 |\mathcal{S}_i|\left(|\mathcal{S}_i|G^2 + |\mathcal{S}_i|\|\nabla f(\theta^{t-1})\|^2\right)\right] \tag{29}$$

$$= 2\mathbb{E}\left[\left(\frac{1}{|\mathcal{S}_\tau^t|} - \frac{1}{|\mathcal{S}|}\right)^2 |\mathcal{S}_i|^2\right]\left(G^2 + \|\nabla f(\theta^{t-1})\|^2\right) \tag{30}$$

Now, note that $\mathcal{S}_\tau^t \subseteq \mathcal{S} \implies |\mathcal{S}_i| = |\mathcal{S}_\tau^t|$. Therefore,

$$\mathcal{T}_3 \leq 2\mathbb{E}\left[\left(\frac{1}{|\mathcal{S}_\tau^t|} - \frac{1}{|\mathcal{S}|}\right)^2 |\mathcal{S}_i|^2\right]\left(G^2 + \|\nabla f(\theta^{t-1})\|^2\right) \tag{31}$$

$$= 2\mathbb{E}\left[\left(\frac{|\mathcal{S}| - |\mathcal{S}_\tau^t|}{|\mathcal{S}|}\right)^2\right]\left(G^2 + \|\nabla f(\theta^{t-1})\|^2\right) \tag{32}$$

Moving now to $\mathcal{T}_4$, we have:

$$\mathcal{T}_4 = \mathbb{E}\left[\left\|\sum_{k\in\mathcal{S}_d}\frac{1}{|\mathcal{S}|}g_k^t\right\|^2\right] \leq \mathbb{E}\left[\left(\frac{1}{|\mathcal{S}|}\right)^2\left\|\sum_{k\in\mathcal{S}_d}g_k^t\right\|^2\right] \tag{33}$$

$$\overset{\text{lemma B.3}}{\leq} \mathbb{E}\left[\left(\frac{1}{|\mathcal{S}|}\right)^2|\mathcal{S}_d|\sum_{k\in\mathcal{S}_d}\|g_k^t\|^2\right] \tag{34}$$

$$= \mathbb{E}\left[\left(\frac{1}{|\mathcal{S}|}\right)^2|\mathcal{S}_d|\sum_{k\in\mathcal{S}_d}\left\|g_k^t - \nabla f(\theta^{t-1}) + \nabla f(\theta^{t-1})\right\|^2\right] \tag{35}$$

$$\overset{\text{lemma B.3}}{\leq} 2\mathbb{E}\left[\left(\frac{1}{|\mathcal{S}|}\right)^2|\mathcal{S}_d|\sum_{k\in\mathcal{S}_d}\left(\left\|g_k^t - \nabla f(\theta^{t-1})\right\|^2 + \left\|\nabla f(\theta^{t-1})\right\|^2\right)\right] \tag{36}$$

$$\overset{\text{assumption 4.3}}{\leq} 2\mathbb{E}\left[\left(\frac{1}{|\mathcal{S}|}\right)^2|\mathcal{S}_d|\left(|\mathcal{S}_d|G^2 + \sum_{k\in\mathcal{S}_d}\left\|\nabla f(\theta^{t-1})\right\|^2\right)\right] \tag{37}$$

$$= 2\mathbb{E}\left[\left(\frac{1}{|\mathcal{S}|}\right)^2|\mathcal{S}_d|\left(|\mathcal{S}_d|G^2 + |\mathcal{S}_d|\left\|\nabla f(\theta^{t-1})\right\|^2\right)\right] \tag{38}$$

$$= 2\mathbb{E}\left[\left(\frac{|\mathcal{S}_d|}{|\mathcal{S}|}\right)^2\right]\left(G^2 + \left\|\nabla f(\theta^{t-1})\right\|^2\right) \tag{39}$$

$$\tag{40}$$

Observing that $|\mathcal{S}_d| = |\mathcal{S}| - |\mathcal{S}_\tau^t|$ we obtain:

$$\mathcal{T}_4 \leq 2\mathbb{E}\left[\left(\frac{|\mathcal{S}_d|}{|\mathcal{S}|}\right)^2\right]\left(G^2 + \left\|\nabla f(\theta^{t-1})\right\|^2\right) = \mathbb{E}\left[\left(\frac{|\mathcal{S}| - |\mathcal{S}_\tau^t|}{|\mathcal{S}|}\right)^2\right]\left(G^2 + \left\|\nabla f(\theta^{t-1})\right\|^2\right) \tag{41}$$

Finally, by plugging (31) and (41) in (22) we obtain

$$\mathbb{E}_{\mathcal{S}^t\sim\mathcal{U}(\mathcal{S})}\left[\left\|g^{(t)\tau}(\theta) - \nabla f(\theta)\right\|^2\right] \leq 8\mathbb{E}_{\mathcal{S}^t\sim\mathcal{U}(\mathcal{S})}\left[\left(\frac{|\mathcal{S}| - |\mathcal{S}_\tau^t|}{|\mathcal{S}|}\right)^2\right]\left(G^2 + \|\nabla f(\theta)\|^2\right)$$

which concludes the proof.

$\square$

**Proof of Corollary 4.8** This corollary follows from Lemma 4.7, which states that

$$\mathbb{E}_{\mathcal{S}^t\sim\mathcal{U}(\mathcal{S})}\left[\left\|g^{(t)\tau}(\theta) - \nabla f(\theta)\right\|^2\right] \leq 8\mathbb{E}_{\mathcal{S}^t\sim\mathcal{U}(\mathcal{S})}\left[\left(\frac{|\mathcal{S}| - |\mathcal{S}_\tau^t|}{|\mathcal{S}|}\right)^2\right]\left(G^2 + \|\nabla f(\theta)\|^2\right)$$

To prove the results, we use (i) Assumption 4.4, (ii) the fact that $|\mathcal{S}^t| = |\mathcal{S}|C \,\forall t$ and (iii) $\mathcal{S}_\tau^t$ is union of $\tau$ disjoint $\mathcal{S}^t$ sets. Using points (i)-(iii), and assuming $\tau \in [0, \frac{1}{C}]$, it follows that:

$$\left\|g^{(t)\tau}(\theta) - \nabla f(\theta)\right\|^2 \leq 8\left(1 - \tau C\right)^2\left(G^2 + \|\nabla f(\theta)\|^2\right)$$

$\square$

**Proof of Lemma 4.10** (Bounded error of delayed gradients)

Note that, by Assumption 4.4, $|\mathcal{S}^t| = |\mathcal{S}|C \,\forall t$, and that $|\mathcal{S}|C\tau = |\mathcal{S}_\tau^t|$:

$$\Lambda_t = \mathbb{E}\left[\left\|\frac{1}{\tau}\sum_{k=t-\tau+1}^{t}\frac{1}{|\mathcal{S}^k|J}\sum_{i=1}^{|\mathcal{S}^k|}\sum_{j=1}^{J}\tilde{g}_i^{k,j}(\theta_i^{k,j-1}) - g^{t_\tau}\right\|^2\right] \tag{42}$$

$$= \mathbb{E}\left[\left\|\frac{1}{\tau}\sum_{k=t-\tau+1}^{t}\frac{1}{|\mathcal{S}^k|J}\sum_{i=1}^{|\mathcal{S}^k|}\sum_{j=1}^{J}\left(\tilde{g}_i^{k,j}(\theta_i^{k,j-1}) - g_i(\theta^{t-1})\right)\right\|^2\right] \tag{43}$$

$$= \mathbb{E}\left[\left\|\frac{1}{\tau}\sum_{k=t-\tau+1}^{t}\frac{1}{|\mathcal{S}^k|J}\sum_{i=1}^{|\mathcal{S}^k|}\sum_{j=1}^{J}\left(\tilde{g}_i^{k,j}(\theta_i^{k,j-1}) - g_i(\theta_i^{k,j-1}) + g_i(\theta_i^{k,j-1}) - g_i(\theta^{k-1}) + g_i(\theta^{k-1}) - g_i(\theta^{t-1})\right)\right\|^2\right] \tag{44}$$

$$\leq 3\left(\mathcal{T}_1 + \mathcal{T}_2 + \mathcal{T}_3\right) \tag{45}$$

$$\mathcal{T}_1 = \mathbb{E}\left[\left\|\frac{1}{\tau}\sum_{k=t-\tau+1}^{t}\frac{1}{|\mathcal{S}^k|J}\sum_{i=1}^{|\mathcal{S}^k|}\sum_{j=1}^{J}\left(\tilde{g}_i^{k,j}(\theta_i^{k,j-1}) - g_i(\theta_i^{k,j-1})\right)\right\|^2\right] \tag{46}$$

$$\leq \frac{1}{\tau}\frac{\sigma^2}{|\mathcal{S}^t|J} = \frac{\sigma^2}{|\mathcal{S}_\tau^t|J} \tag{47}$$

$$\mathcal{T}_2 = \mathbb{E}\left[\left\|\frac{1}{\tau}\sum_{k=t-\tau+1}^{t}\frac{1}{|\mathcal{S}^k|J}\sum_{i=1}^{|\mathcal{S}^k|}\sum_{j=1}^{J}\left(g_i(\theta_i^{k,j-1}) - g_i(\theta^{k-1})\right)\right\|^2\right] \tag{48}$$

$$\leq \frac{L^2}{|\mathcal{S}|J\tau}\sum_{k=t-\tau+1}^{t}\sum_{i=1}^{|\mathcal{S}|}\sum_{j=1}^{J}\mathbb{E}\left[\left\|\theta^{k,j-1} - \theta^{k-1}\right\|^2\right] \tag{49}$$

$$= \frac{L^2}{\tau}\sum_{k=t-\tau+1}^{t}\mathcal{U}_k \tag{50}$$

$$\mathcal{T}_3 = \mathbb{E}\left[\left\|\frac{1}{\tau}\sum_{k=t-\tau+1}^{t}\frac{1}{|\mathcal{S}^k|J}\sum_{i=1}^{|\mathcal{S}^k|}\sum_{j=1}^{J}\left(g_i(\theta^{k-1}) - g_i(\theta^{t-1})\right)\right\|^2\right] \tag{51}$$

$$\leq \frac{L^2}{|\mathcal{S}|\tau}\sum_{k=t-\tau+1}^{t}\sum_{i=1}^{|\mathcal{S}|}\mathbb{E}\left[\left\|\theta^{k-1} - \theta^{t-1}\right\|^2\right] \tag{52}$$

$$\leq \frac{L^2}{\tau}\sum_{k=t-\tau+1}^{t}\mathbb{E}\left[\left\|\theta^{k-1} - \theta^{t-1}\right\|^2\right] \tag{53}$$

$$= \frac{L^2}{\tau}\sum_{k=t-\tau+1}^{t}(t-k)\,\mathbb{E}\left[\left\|\theta^{k} - \theta^{k-1}\right\|^2\right] \tag{54}$$

$$\leq 2L^2\eta^2\sum_{k=t-\tau+1}^{t-1}\left(\mathbb{E}\left[\left\|\nabla f(\theta^{k-1})\right\|^2\right] + \mathcal{E}_k\right) \tag{55}$$

So, combining with lemma Lemmas B.5 and B.6 we have:

$$\sum_{t=1}^{T} \Lambda_t \leq 3 \left( \frac{T\sigma^2}{|\mathcal{S}_\tau^t|J} + L^2 \sum_{t=1}^{T} \mathcal{U}_t + 2L^2 \eta^2 (\tau - 1) \sum_{t=1}^{T-1} \left( \mathbb{E}\left[ \left\| \nabla f(\theta^{t-1}) \right\|^2 \right] + \mathcal{E}_t \right) \right) \tag{56}$$

$$\overset{\text{lemma B.5}}{=} 3\left( \frac{T\sigma^2}{|\mathcal{S}_\tau^t|J} + 2L^2 \eta^2 (\tau - 1) \sum_{t=1}^{T-1} \left( \mathbb{E}\left[ \left\| \nabla f(\theta^{t-1}) \right\|^2 \right] + \mathcal{E}_t \right) \right. \tag{57}$$

$$\left. + \underbrace{L^2 T J \eta_l^2 \beta^2 \sigma^2 \left( 1 + 2J^3 \eta_l^2 \beta^2 L^2 \right)}_{\mathcal{T}_4} + 2J^2 L^2 e^2 \sum_{t=1}^{T} \Xi_t \right) \right)$$

$$\overset{\text{lemma B.6}}{=} 3\left( \frac{T\sigma^2}{|\mathcal{S}_\tau^t|J} + 2L^2 \eta^2 (\tau - 1) \sum_{t=1}^{T-1} \left( \mathbb{E}\left[ \left\| \nabla f(\theta^{t-1}) \right\|^2 \right] + \mathcal{E}_t \right) \right. \tag{58}$$

$$+ \mathcal{T}_4 + \underbrace{2J^2 L^2 e^2 \left( 4\eta_l^2 \left( (1-\beta)^2 + e(\beta \eta L T)^2 \right) \right)}_{\alpha_1} \sum_{t=0}^{T-1} \left( \mathcal{E}_t + \mathbb{E}\left[ \left\| \nabla f(\theta^{t-1}) \right\|^2 \right] \right)$$

$$\left. + \underbrace{2e^2 J^2 L^2 (2e\eta_l^2 \beta \tau T G_\tau)}_{\mathcal{T}_5} \right)$$

$$= 3\left( \frac{T\sigma^2}{|\mathcal{S}_\tau^t|J} + \mathcal{T}_4 + \underbrace{(\alpha_1 + 2L^2 \eta_l^2 (\tau - 1))}_{\alpha_2} \sum_{t=1}^{T-1} \left( \mathbb{E}\left[ \left\| \nabla f(\theta^{t-1}) \right\|^2 \right] + \mathcal{E}_t \right) + \mathcal{T}_5 \right) \tag{59}$$

$$\square$$

## B.4 Convergence Proof

**Lemma B.4** (Bounded variance of server updates)**.** *Under Assumptions 4.1 and 4.2, it holds that:*

$$\sum_{t=1}^{T} \mathcal{E}_t \leq \frac{8}{5\beta} \mathcal{E}_0 + \frac{3}{5} \sum_{t=0}^{T-1} \mathbb{E}\left[ \left\| \nabla f(\theta^{t-1}) \right\|^2 \right] + 21\beta \frac{\sigma^2}{|\mathcal{S}_\tau^t|J} T + \tag{60}$$

$$+ \frac{448}{5} (\eta_l J L)^2 (e^3 \tau T) G_\tau + 6\beta \sum_{t=1}^{T} \gamma_t$$

*Proof.*

$$\mathcal{E}_t := \mathbb{E}\left[ \left\| \nabla f(\theta^{t-1}) - \tilde{m}_\tau^{t+1} \right\|^2 \right] \tag{61}$$

$$= \mathbb{E}\left[ \left\| (1-\beta)(\nabla f(\theta^{t-1}) - \tilde{m}_\tau^t) + \beta(\nabla f(\theta^{t-1}) - \tilde{g}^{t_\tau}) \right\|^2 \right] \tag{62}$$

$$= \mathbb{E}\left[ \left\| (1-\beta)(\nabla f(\theta^{t-1}) - \tilde{m}_\tau^t) \right\|^2 \right] + \beta^2 \mathbb{E}\left[ \left\| (\nabla f(\theta^{t-1}) - \tilde{g}^{t_\tau}) \right\|^2 \right] \tag{63}$$

$$+ 2\beta \mathbb{E}\left[ \left\langle (1-\beta)(\nabla f(\theta^{t-1}) - \tilde{m}_\tau^t), \nabla f(\theta^{t-1}) - \frac{1}{\tau} \sum_{k=t-\tau+1}^{t} \frac{1}{|\mathcal{S}^k|J} \sum_{i=1}^{|\mathcal{S}^k|} \sum_{j=1}^{J} g_i(\theta_i^{k,j-1}) \right\rangle \right] \tag{64}$$

Using the AM-GM inequality and Lemma B.3:

$$\leq \left(1 + \frac{\beta}{2}\right) \mathbb{E}\left[\left\|(1-\beta)(\nabla f(\theta^{t-1}) - \tilde{m}_\tau^t)\right\|^2\right] + 2\beta^2\left(\gamma_t + \Lambda_t\right) +$$

$$+ 4\beta\gamma_t + 8\beta \left(\frac{L^2}{\tau}\sum_{k=t-\tau+1}^{t}\mathcal{U}_k + 2L^2\eta^2\sum_{k=t-\tau+1}^{t-1}\left(\mathbb{E}\left[\left\|\nabla f(\theta^{k-1})\right\|^2\right] + \mathcal{E}_k\right)\right) \tag{65}$$

$$\overset{\text{lemma 4.10}}{\leq} \left(1 + \frac{\beta}{2}\right) \mathbb{E}\left[\left\|(1-\beta)(\nabla f(\theta^{t-1}) - \tilde{m}_\tau^t)\right\|^2\right] + \left(2\beta^2 + 4\beta\right)\gamma_t + 6\beta^2\frac{\sigma^2}{|\mathcal{S}_\tau^t|J} + \tag{66}$$

$$+ \left(6\beta^2 + 8\beta\right)\underbrace{\left(\frac{L^2}{\tau}\sum_{k=t-\tau+1}^{t}\mathcal{U}_k + 2L^2\eta^2\sum_{k=t-\tau+1}^{t-1}\left(\mathbb{E}\left[\left\|\nabla f(\theta^{k-1})\right\|^2\right] + \mathcal{E}_k\right)\right)}_{\mathcal{T}_1}$$

$$\leq (1-\beta)^2\left(1 + \frac{\beta}{2}\right)\mathbb{E}\left[\left\|\nabla f(\theta^{t-2}) - \tilde{m}_\tau^t + \nabla f(\theta^{t-1}) - \nabla f(\theta^{t-2})\right\|^2\right] + \tag{67}$$

$$+ 6\beta^2\frac{\sigma^2}{|\mathcal{S}_\tau^t|J} + 6\beta\gamma_t + 14\beta\mathcal{T}_1$$

Applying the AM-GM inequality again:

$$\leq (1-\beta)^2\left(1 + \frac{\beta}{2}\right)\left[\left(1 + \frac{\beta}{4}\right)\mathbb{E}\left[\left\|\nabla f(\theta^{t-2}) - \tilde{m}_\tau^t\right\|^2\right] + \tag{68}$$

$$+ \left(1 + \frac{1}{\beta}\right)\mathbb{E}\left[\left\|\nabla f(\theta^{t-1}) - \nabla f(\theta^{t-2})\right\|^2\right]\right] + 6\beta^2\frac{\sigma^2}{|\mathcal{S}_\tau^t|J} + 6\beta\gamma_t + 14\beta\mathcal{T}_1$$

$$\overset{\text{assumption 4.2}}{\leq} (1-\beta)^2\left(1 + \frac{\beta}{2}\right)\left[\left(1 + \frac{\beta}{4}\right)\mathcal{E}_{t-1} + \tag{69}$$

$$+ \left(1 + \frac{1}{\beta}\right)L^2\mathbb{E}\left[\left\|\theta^{t-1} - \theta^{t-2}\right\|^2\right]\right] + 6\beta^2\frac{\sigma^2}{|\mathcal{S}_\tau^t|J} + 6\beta\gamma_t + 14\beta\mathcal{T}_1$$

$$\leq (1-\beta)^2\left(1 + \frac{\beta}{2}\right)\left[\left(1 + \frac{\beta}{4}\right)\mathcal{E}_{t-1} + \tag{70}$$

$$+ 2\left(1 + \frac{1}{\beta}\right)L^2\eta^2\left(\mathbb{E}\left[\left\|\nabla f(\theta^{t-2})\right\|^2\right] + \mathcal{E}_{t-1}\right)\right] + 6\beta^2\frac{\sigma^2}{|\mathcal{S}_\tau^t|J} + 6\beta\gamma_t + 14\beta\mathcal{T}_1$$

Where in the last inequality we used the fact that:

$$\left\|\theta^{t-1} - \theta^{t-2}\right\|^2 \leq 2\eta^2\left(\left\|\nabla f(\theta^{t-2})\right\|^2 + \left\|\nabla f(\theta^{t-2}) - \tilde{m}_\tau^t\right\|^2\right).$$

Now notice that $(1-\beta)^2\left(1 + \frac{\beta}{2}\right)\left(1 + \frac{\beta}{4}\right) \leq (1-\beta)$ and that $2(1-\beta)^2\left(1 + \frac{\beta}{2}\right)\left(1 + \frac{1}{\beta}\right) \leq \frac{2}{\beta}$:

$$\mathcal{E}_t \leq (1-\beta)\mathcal{E}_{t-1} + \frac{2}{\beta}L^2\eta^2\left(\mathbb{E}\left[\left\|\nabla f(\theta^{t-2})\right\|^2\right] + \mathcal{E}_{t-1}\right) + 6\beta^2\frac{\sigma^2}{|\mathcal{S}_\tau^t|J} + 6\beta\gamma_t + 14\beta\mathcal{T}_1 \tag{71}$$

$$= \left(1 - \beta + \frac{2}{\beta}L^2\eta^2\right)\mathcal{E}_{t-1} + \frac{2}{\beta}L^2\eta^2\mathbb{E}\left[\left\|\nabla f(\theta^{t-2})\right\|^2\right] + 6\beta^2\frac{\sigma^2}{|\mathcal{S}_\tau^t|J} + 6\beta\gamma_t + 14\beta\mathcal{T}_1 \tag{72}$$

Define:

- $\mathcal{T}_2 := L^2 T J \eta_l^2 \beta^2 \sigma^2\left(1 + 2J^3\eta_l^2\beta^2 L^2\right)$

- $\mathcal{T}_3 := 2e^2 J^2 L^2(2e\eta_l^2\beta\tau T G_\tau)$

- $\alpha_1 := 2J^2 L^2 e^2\left(4\eta_l^2\left((1-\beta)^2 + e(\beta\eta L T)^2\right)\right) + 2L^2\eta_l^2(\tau - 1)$

Summing up over $T$ and substituting into $\mathcal{T}_1$ the expression for $\mathcal{U}_t$:

$$\sum_{t=1}^{T} \mathcal{E}_t \leq \underbrace{\left(1 - \beta + \frac{2}{\beta}L^2\eta^2 + 14\beta\alpha_1\right)}_{\alpha_2} \sum_{t=0}^{T-1} \mathcal{E}_t + \tag{73}$$

$$+ \underbrace{\left(\frac{2}{\beta}L^2\eta^2 + 14\beta\alpha_1\right)}_{\alpha_3} \sum_{t=0}^{T-1} \mathbb{E}\left[\left\|\nabla f(\theta^{t-1})\right\|^2\right] +$$

$$+ 14\beta\left(\mathcal{T}_2 + \mathcal{T}_3\right)T + 6\beta^2\frac{\sigma^2}{|\mathcal{S}_\tau^t|J}T + 6\beta\sum_{t=1}^{T}\gamma_t$$

We now have that:

$$\alpha_2 := \left(1 - \beta + \frac{2}{\beta}L^2\eta^2 + 14\beta\left[2J^2L^2e^2\left(4\eta_l^2\left((1-\beta)^2 + e(\beta\eta LT)^2\right)\right) + 2L^2\eta_l^2(\tau-1)\right]\right) \tag{74}$$

$$= \left(1 - \beta + \frac{2}{\beta}L^2\eta^2 + 14\beta\left[8J^2L^2e^2\eta_l^2\left((1-\beta)^2 + e(\beta\eta LT)^2\right) + 2L^2\eta_l^2(\tau-1)\right]\right) \tag{75}$$

$$\leq \left(1 - \beta + \frac{2}{\beta}L^2\eta^2 + 112\beta e^2(\eta_l JL)^2\left[(1-\beta)^2 + (\beta\eta LT)^2 + (\tau-1)\right]\right) \tag{76}$$

$$\tag{77}$$

Now impose $(\eta_l JL) \leq (37\sqrt{\tau}\beta\eta LTe)^{-1}$ and $\eta \leq \frac{\beta}{\sqrt{8}L}$. We have that:

$$\alpha_2 \leq \left(1 - \beta + \frac{2\beta}{8} + \frac{\beta}{8}\right) = \left(1 - \frac{5\beta}{8}\right) \tag{78}$$

$$\alpha_3 \leq \frac{3\beta}{8} \tag{79}$$

$$14\beta\mathcal{T}_2 = 14\beta L^2 TJ\eta_l^2\beta^2\sigma^2\left(1 + 2J^3\eta_l^2\beta^2L^2\right) \tag{80}$$

$$= 14\beta^3(\eta_l JL)^2\left(\frac{1}{J} + 2(\eta_l JL\beta)^2\right)\sigma^2 T \tag{81}$$

$$\leq 7\beta^2\frac{\sigma^2}{|\mathcal{S}_\tau^t|J}T \tag{82}$$

Where in the last inequality we apply:

$$2\beta(\eta_l JL)^2\left(\frac{1}{J} + 2(\eta_l JL\beta)^2\right) \leq \frac{1}{|\mathcal{S}_\tau^t|J}$$

Plugging all the terms together we have:

$$\sum_{t=1}^{T}\mathcal{E}_t \leq \left(1 - \frac{5}{8\beta}\right)\sum_{t=0}^{T-1}\mathcal{E}_t + \frac{3\beta}{8}\sum_{t=0}^{T-1}\mathbb{E}\left[\left\|\nabla f(\theta^{t-1})\right\|^2\right] + 13\beta^2\frac{\sigma^2}{|\mathcal{S}_\tau^t|J}T + \tag{83}$$

$$+ 56\beta(\eta_l JL)^2(e^3\tau T)G_\tau + 6\beta\sum_{t=1}^{T}\gamma_t$$

Rearranging the terms completes the proof. $\qquad\square$

**Lemma B.5.** *Under Assumptions 4.1 and 4.2, for Eq. (9) it holds that:*

$$\mathcal{U}_t \leq 2J^2e^2\Xi_t + J\eta_l^2\beta^2\sigma^2(1 + 2J^3\eta_l^2L^2\beta^2) \tag{84}$$

$$\sum_{t=1}^{T}\mathcal{U}_t \leq TJ\eta_l^2\beta^2\sigma^2(1 + 2J^3\eta_l^2\beta^2L^2) + 2J^2e^2\sum_{t=1}^{T}\Xi_t \tag{85}$$

*Proof.*

$$\mathbb{E}\left[\left\|\theta_i^{t,j} - \theta^{t-1}\right\|^2\right] \leq 2\mathbb{E}\left[\left\|\sum_{k=0}^{j-1} \zeta_i^{t,k}\right\|^2\right] + 2j\eta_l^2\beta^2\sigma^2 \tag{86}$$

$$\stackrel{\text{lemma B.3}}{\leq} 2j\sum_{k=0}^{j-1}\mathbb{E}\left[\left\|\zeta_i^{t,k}\right\|^2\right] + 2j\eta_l^2\beta^2\sigma^2 \tag{87}$$

For any $1 \leq k \leq j-1 \leq J-2$, using $\eta L \leq \frac{1}{\beta J} \leq \frac{1}{\beta(j+1)}$, we have:

$$\mathbb{E}\left[\left\|\zeta_i^{t,k}\right\|^2\right] \leq \left(1 + \frac{1}{j}\right)\mathbb{E}\left[\left\|\zeta_i^{t,k-1}\right\|^2\right] + (1+j)\mathbb{E}\left[\left\|\zeta_i^{t,k} - \zeta_i^{t,k-1}\right\|^2\right] \tag{88}$$

$$\leq \left(1 + \frac{1}{j}\right)\mathbb{E}\left[\left\|\zeta_i^{t,k-1}\right\|^2\right] + (1+j)\eta_l^2\beta^2 L^2\left(\eta_l^2\beta^2\sigma^2 + \mathbb{E}\left[\left\|\zeta_i^{t,k-1}\right\|^2\right]\right) \tag{89}$$

$$\leq \left(1 + \frac{1}{j}\right)\mathbb{E}\left[\left\|\zeta_i^{t,k-1}\right\|^2\right] + (1+j)\eta_l^4\beta^4 L^2\sigma^2 + \frac{1}{1+j}\mathbb{E}\left[\left\|\zeta_i^{t,k} - \zeta_i^{t,k-1}\right\|^2\right] \tag{90}$$

$$\leq \left(1 + \frac{2}{j}\right)\mathbb{E}\left[\left\|\zeta_i^{t,k-1}\right\|^2\right] + (1+j)\eta_l^4\beta^4 L^2\sigma^2 \tag{91}$$

$$\stackrel{\left(1+\frac{2}{j}\right)^j \leq e^2}{\leq} e^2\mathbb{E}\left[\left\|\zeta_i^{t,0}\right\|^2\right] + 4j^2\eta_l^4\beta^4 L^2\sigma^2 \tag{92}$$

So it holds that:

$$\mathbb{E}\left[\left\|\theta_i^{t,j} - \theta^{t-1}\right\|^2\right] \leq 2j^2\left(e^2\mathbb{E}\left[\left\|\zeta_i^{t,0}\right\|^2\right] + 4j^2\eta_l^4 L^2\sigma^2\right) + 2j\eta_l^2\sigma^2 \tag{93}$$

$$= 2e^2 j^2\mathbb{E}\left[\left\|\zeta_i^{t,0}\right\|^2\right] + 2j\eta_l^2\sigma^2\beta^2(1 + 4j^3\eta_l^2 L^2\beta^2) \tag{94}$$

So, summing up over $i$ and $j$:

$$\mathcal{U}_t \leq \frac{1}{|\mathcal{S}|J}\sum_{i=1}^{|\mathcal{S}|}\sum_{j=1}^{J} 2e^2 j^2\mathbb{E}\left[\left\|\zeta_i^{t,0}\right\|^2\right] + 2j\eta_l^2\sigma^2\beta^2(1 + 4j^3\eta_l^2 L^2\beta^2) \tag{95}$$

$$\leq 2J^2 e^2\Xi_t + J\eta_l^2\beta^2\sigma^2(1 + 2J^3\eta_l^2 L^2\beta^2) \tag{96}$$

Finally, summing up over $T$:

$$\sum_{t=1}^{T}\mathcal{U}_t \leq \underbrace{TJ\eta_l^2\beta^2\sigma^2(1 + 2J^3\eta_l^2\beta^2 L^2)}_{\mathcal{T}_1} + 2J^2 e^2\sum_{t=1}^{T}\Xi_t \tag{97}$$

$$\leq \mathcal{T}_1 + 2J^2 e^2\left(4\eta^2\left((1-\beta)^2 + e(\beta\eta LT)^2\right)\sum_{t=1}^{T-1}\left(\mathcal{E}_t + \mathbb{E}\left[\left\|\nabla f(\theta^{t-1})\right\|^2\right]\right) + \underbrace{2e\eta^2\beta^2\tau TG_\tau}_{\mathcal{T}_2}\right) \tag{98}$$

$$\leq \mathcal{T}_1 + \alpha_1\sum_{t=1}^{T-1}\left(\mathcal{E}_t + \mathbb{E}\left[\left\|\nabla f(\theta^{t-1})\right\|^2\right]\right) + \alpha_2\mathcal{T}_2 \tag{99}$$

$\square$

**Lemma B.6.** *Under Assumptions 4.1, 4.2 and 4.4, if $224e(\eta_l JL)^2\left((1-\beta)^2 + e(\beta\eta LT)^2\right) \leq 1$, for Eq. (11) it holds for $t \geq 0$ that:*

$$\Xi_t \leq \frac{1}{56eJ^2 L^2}\sum_{t=0}^{T-1}\left(\mathcal{E}_t + \mathbb{E}\left[\left\|\nabla f(\theta^{t-1})\right\|^2\right]\right) + 2e\eta_l^2\beta^2\tau TG_\tau \tag{100}$$

*Proof.* Note that $\zeta_i^{t,0} = -\eta_l \left( (1-\beta)\tilde{m}_\tau^t + \beta g_i(\theta^{t-1}) \right)$,

$$\frac{1}{|\mathcal{S}|} \sum_{i=1}^{|\mathcal{S}|} \left\| \zeta_i^{t,0} \right\|^2 \leq 2\eta_l^2 \left( (1-\beta)^2 \left\| \tilde{m}_\tau^t \right\|^2 + \frac{\beta^2}{|\mathcal{S}|} \sum_{i=1}^{|\mathcal{S}|} \left\| g_i(\theta^{t-1}) \right\|^2 \right) \tag{101}$$

For any $a > 0$, considering each client participates to the train every $\tau = \frac{1}{C}$ rounds:

$$\mathbb{E}\left[ \left\| g_i(\theta^{t-1}) \right\|^2 \right] = \mathbb{E}\left[ \left\| g_i(\theta^{t-1}) - g_i(\theta^{t-\tau-1}) + g_i(\theta^{t-\tau-1}) \right\|^2 \right] \tag{102}$$

$$\overset{\text{lemma B.3}}{\leq} (1+a)\mathbb{E}\left[ \left\| g_i(\theta^{t-\tau-1}) \right\|^2 \right] + \tag{103}$$

$$+ \left(1 + \frac{1}{a}\right) \mathbb{E}\left[ \left\| g_i(\theta^{t-1}) - g_i(\theta^{t-\tau-1}) \right\|^2 \right] \tag{}$$

$$\leq (1+a)\mathbb{E}\left[ \left\| g_i(\theta^{t-\tau-1}) \right\|^2 \right] + \tag{104}$$

$$+ \left(1 + \frac{1}{a}\right) L^2 \mathbb{E}\left[ \left\| \theta^{t-1} - \theta^{t-\tau-1} \right\|^2 \right] \tag{105}$$

$$\leq (1+a)\mathbb{E}\left[ \left\| g_i(\theta^{t-\tau-1}) \right\|^2 \right] + \tag{106}$$

$$+ 2\left(1 + \frac{1}{a}\right) L^2 \eta^2 \tau \sum_{k=1}^{\tau} \left( \mathcal{E}_{t-k} + \mathbb{E}\left[ \left\| \nabla f(\theta^{t-k-1}) \right\|^2 \right] \right) \tag{107}$$

$$\leq (1+a)^{\frac{t}{\tau}} \mathbb{E}\left[ \left\| g_i(\theta^{t_i-1}) \right\|^2 \right] + \tag{108}$$

$$+ 2\left(1 + \frac{1}{a}\right) L^2 \eta^2 \tau \sum_{s=1}^{\frac{t}{\tau}} \sum_{k=1}^{\tau} \left( \mathcal{E}_{s\tau-k} + \mathbb{E}\left[ \left\| \nabla f(\theta^{s\tau-k}) \right\|^2 \right] \right) (1+a)^{\frac{t}{\tau}-s} \tag{}$$

$$\leq (1+a)^{\frac{t}{\tau}} \mathbb{E}\left[ \left\| g_i(\theta^{t_i-1}) \right\|^2 \right] + \tag{109}$$

$$+ 2\left(1 + \frac{1}{a}\right) L^2 \eta^2 \tau \sum_{k=1}^{t-1} \left( \mathcal{E}_k + \mathbb{E}\left[ \left\| \nabla f(\theta^{k-1}) \right\|^2 \right] \right) (1+a)^{\frac{t}{\tau}} \tag{}$$

Where $t_i := \min_{t \in [T]}(t \text{ s.t. } i \in \mathcal{S}^t)$. Now take $a = \frac{\tau}{t}$:

$$\mathbb{E}\left[ \left\| g_i(\theta^{t-1}) \right\|^2 \right] \leq e\mathbb{E}\left[ \left\| g_i(\theta^{t_i-1}) \right\|^2 \right] + \tag{110}$$

$$+ 2e\eta^2 L^2 \tau \left(\frac{t}{\tau} + 1\right) \sum_{k=1}^{t-1} \left( \mathcal{E}_k + \mathbb{E}\left[ \left\| \nabla f(\theta^{k-1}) \right\|^2 \right] \right) \tag{}$$

So:

$$\sum_{t=1}^{T} \Xi_t \leq \sum_{t=1}^{T} 2\eta_l^2 \left( 2(1-\beta)^2 \left( \mathcal{E}_{t-1} + \mathbb{E}\left[ \left\| \nabla f(\theta^{t-2}) \right\|^2 \right] \right) + \frac{\beta^2}{|\mathcal{S}|} \sum_{i=1}^{|\mathcal{S}|} \mathbb{E}\left[ \left\| g_i(\theta^{t-1}) \right\|^2 \right] \right) \tag{111}$$

$$\leq \sum_{t=1}^{T} 4\eta_l^2 (1-\beta)^2 \left( \mathcal{E}_{t-1} + \mathbb{E}\left[ \left\| \nabla f(\theta^{t-2}) \right\|^2 \right] \right) + \tag{112}$$

$$+ 2\eta_l^2 \beta^2 \sum_{t=1}^{T} \left( \frac{e}{|\mathcal{S}|} \sum_{i=1}^{|\mathcal{S}|} \mathbb{E}\left[ \left\| g_i(\theta^{t_i-1}) \right\|^2 \right] + 2e\eta_l^2 L^2 \tau \left(\frac{t}{\tau} + 1\right) \sum_{k=1}^{t-1} \left( \mathcal{E}_k + \mathbb{E}\left[ \left\| \nabla f(\theta^{t-1}) \right\|^2 \right] \right) \right) \tag{}$$

$$\leq 4\eta_l^2 (1-\beta)^2 \sum_{t=1}^{T} \left( \mathcal{E}_{t-1} + \mathbb{E}\left[ \left\| \nabla f(\theta^{t-2}) \right\|^2 \right] \right) + \tag{113}$$

$$+ 2\eta_l^2 \beta^2 \left( eT \sum_{t=1}^{\tau} G_t + 2e(\eta L T)^2 \sum_{t=1}^{T-1} \left( \mathcal{E}_t + \mathbb{E}\left[ \left\| \nabla f(\theta^{t-1}) \right\|^2 \right] \right) \right) \tag{}$$

Let us define $G_\tau := \max_{t \in [1,\tau]} G_t$, with $G_t := \frac{1}{|\mathcal{S}^t|} \sum_{i=1}^{|\mathcal{S}^t|} \mathbb{E}\left[\left\|g_i(\theta^{t-1})\right\|^2\right]$. We have that:

$$\sum_{t=1}^{T} \Xi_t \leq 4\eta_l^2 \left((1-\beta)^2 + e(\beta\eta LT)^2\right) \sum_{t=0}^{T-1} \left(\mathcal{E}_t + \mathbb{E}\left[\left\|\nabla f(\theta^{t-1})\right\|^2\right]\right) + 2e\eta_l^2\beta^2\tau T G_\tau \tag{114}$$

Applying the upper bound of $\eta_l$ completes the proof. $\qquad\square$

**Lemma B.7** (Cheng et al. (2024)). *Under Assumption 4.2, if $\eta L \leq \frac{1}{24}$, the following holds for all $t \geq 0$:*

$$\mathbb{E}\left[f(\theta^t)\right] \leq \mathbb{E}\left[f(\theta^{t-1})\right] - \frac{11\eta}{24}\mathbb{E}\left[\left\|\nabla f(\theta^{t-1})\right\|^2\right] + \frac{13\eta}{24}\mathcal{E}_t \tag{115}$$

*Proof.* Since f is $L$-smooth, we have:

$$f(\theta^t) \leq f(\theta^{t-1}) + \left\langle \nabla f(\theta^{t-1}), \theta^t - \theta^{t-1}\right\rangle + \frac{L}{2}\left\|\theta^t - \theta^{t-1}\right\|^2 \tag{116}$$

$$= f(\theta^{t-1}) - \eta\left\|\nabla f(\theta^{t-1})\right\|^2 + \eta\left\langle \nabla f(\theta^{t-1}), \nabla f(\theta^{t-1}) - \tilde{m}_\tau^{t+1}\right\rangle + \frac{L\eta^2}{2}\left\|\tilde{m}_\tau^{t+1}\right\|^2 \tag{117}$$

Since $\theta^t = \theta^{t-1} - \eta\tilde{m}_\tau^{t+1}$, using Young's inequality and imposing $\eta L \leq \frac{1}{24}$, we further have:

$$f(\theta^t) \leq f(\theta^{t-1}) - \frac{\eta}{2}\left\|\nabla f(\theta^{t-1})\right\|^2 + \frac{\eta}{2}\left\|\nabla f(\theta^{t-1}) - \tilde{m}_\tau^{t+1}\right\|^2 + \tag{118}$$

$$+ L\eta^2\left(\left\|\nabla f(\theta^{t-1})\right\|^2 + \left\|\nabla f(\theta^{t-1}) - \tilde{m}_\tau^{t+1}\right\|^2\right)$$

$$\leq f(\theta^{t-1}) - \frac{11\eta}{24}\left\|\nabla f(\theta^{t-1})\right\|^2 + \frac{13\eta}{24}\left\|\nabla f(\theta^{t-1}) - \tilde{m}_\tau^{t+1}\right\|^2 \tag{119}$$

$$\square$$

**Proof of Theorem 4.11** (Convergence rate of GHBM for non-convex functions)

*Under Assumptions 4.1, 4.2 and 4.4, if we take:*

$$\tilde{m}_\tau^0 = 0, \qquad \beta = \min\left\{1, \sqrt{\frac{|\mathcal{S}|JL\Delta}{\sigma^2 T}}\right\}, \qquad \eta = \min\left\{\frac{1}{24L}, \frac{\beta}{\sqrt{8}L}\right\} \tag{120}$$

$$\eta_l JL \lesssim \min\left\{1, \frac{1}{\beta\eta L\sqrt{\tau}T}, \sqrt{\frac{L\Delta}{\beta^3\tau G_\tau T}}, \frac{1}{\sqrt{\beta|\mathcal{S}|}}, \left(\frac{1}{\beta^3|\mathcal{S}|J}\right)^{\frac{1}{4}}\right\}$$

*then GHBM with optimal $\tau = \frac{1}{C}$ converges as:*

$$\frac{1}{T}\sum_{t=1}^{T} \mathbb{E}\left[\left\|\nabla f(\theta^{t-1})\right\|^2\right] \lesssim \frac{L\Delta}{T} + \sqrt{\frac{L\Delta\sigma^2}{|\mathcal{S}|JT}} \tag{121}$$

*Proof.* Combining the results of Lemmas B.4 and B.7, we have that:

$$\sum_{t=1}^{T} \left(\mathbb{E}\left[f(\theta^t)\right] - \mathbb{E}\left[f(\theta^{t-1})\right]\right) \leq -\frac{11\eta}{24}\sum_{t=1}^{T}\mathbb{E}\left[\left\|\nabla f(\theta^{t-1})\right\|^2\right] + \frac{13\eta}{24}\sum_{t=1}^{T}\mathcal{E}_t \tag{122}$$

$$\frac{1}{\eta}\mathbb{E}\left[f(\theta^{t-1}) - f(\theta^0)\right] \leq \frac{26}{30\beta}\mathcal{E}_0 - \frac{1}{15}\sum_{t=1}^{T}\mathbb{E}\left[\left\|\nabla f(\theta^{t-1})\right\|^2\right] + 32\beta\frac{\sigma^2}{|\mathcal{S}_\tau^t|J}T + \tag{123}$$

$$+ \frac{448}{5}(\eta_l JL)^2(e^3\tau T)G_\tau + 6\beta\sum_{t=1}^{T}\gamma_t \tag{124}$$

Imposing $\tau = \frac{1}{C}$, by Corollary 4.8 we have that $\gamma_t = 0$ and $\mathcal{S}_\tau^t = \mathcal{S}$ $\forall t$. Also, noticing that $\tilde{m}_\tau^0 = 0$ implies $\mathcal{E}_0 \leq 2L\left(f(\theta^0) - f^*\right) = 2L\Delta$, we have that:

$$\frac{1}{T}\sum_{t=1}^{T}\mathbb{E}\left[\left\|\nabla f(\theta^{t-1})\right\|^2\right] \lesssim \frac{L\Delta}{\eta_l LT} + \frac{\mathcal{E}_0}{\beta T} + (\eta_l JL\beta)^2\tau G_\tau + \beta\frac{\sigma^2}{|\mathcal{S}|J} \tag{125}$$

$$\lesssim \frac{L\Delta}{T} + \frac{2L\Delta}{\beta T} + (\eta_l JL\beta)^2\tau G_\tau + \beta\frac{\sigma^2}{|\mathcal{S}|J} \tag{126}$$

$$\lesssim \frac{L\Delta}{T} + \frac{2L\Delta}{\beta T} + \beta^2\left(\frac{L\Delta}{\beta^3\tau G_\tau T}\right)\tau G_\tau + \beta\frac{\sigma^2}{|\mathcal{S}|J} \tag{127}$$

$$\lesssim \frac{L\Delta}{T} + \frac{L\Delta}{\beta T} + \beta\frac{\sigma^2}{|\mathcal{S}|J} \tag{128}$$

$$\lesssim \frac{L\Delta}{T} + \sqrt{\frac{L\Delta\sigma^2}{|\mathcal{S}|JT}} \tag{129}$$

where the fourth inequality follows from applying the upper bound $\eta_l JL \leq \sqrt{\frac{L\Delta}{\beta^3\tau G_\tau T}}$ on the third term of Eq. (126). $\qquad\square$

## C  Experimental Setting

### C.1  Datasets and Models

**Cifar-10/100.**  We consider CIFAR-10 and CIFAR-100 to experiment with image classification tasks, each one respectively having 10 and 100 classes. For all methods, training images are preprocessed by applying random crops, followed by random horizontal flips. Both training and test images are finally normalized according to their mean and standard deviation. As the main model for experimentation, we used a model similar to LENET-5 as proposed in (Hsu et al., 2020). To further validate our findings, we also employed a RESNET-20 as described in (He et al., 2015), following the implementation provided in (Idelbayev, 2021). Since batch normalization Ioffe & Szegedy (2015) layers have been shown to hamper performance in learning from decentralized data with skewed label distribution (Hsieh et al., 2020), we replaced them with group normalization (Wu & He, 2018), using two groups in each layer. For a fair comparison, we used the same modified network also in centralized training. We report the result of centralized training for reference in Table 5: as per the hyperparameters, we use 64 for the batch size, 0.01 and 0.1 for the learning rate respectively for the LENET and the RESNET-20 and 0.9 for momentum. We trained both models on both datasets for 150 epochs using a cosine annealing learning rate scheduler.

**Shakespeare.**  The Shakespeare language modeling dataset is created by collating the collective works of William Shakespeare and originally comprises 715 clients, with each client denoting a speaking role. However, for this study, a different approach was used, adopting the LEAF (Caldas et al., 2019) framework to split the dataset among 100 devices and restrict the number of data points per device to 2000. The non-IID dataset is formed by assigning each device to a specific role, and the local dataset for each device contains the sentences from that role. Conversely, the IID dataset is created by randomly distributing sentences from all roles across the devices.

Table 5: **Test accuracy (%) of centralized training over datasets and models used.** Results are reported in term of mean top-1 accuracy over the last 10 epochs, averaged over 5 independent runs.

| DATASET | ACC. CENTRALIZED (%) |
|---|---|
| CIFAR-10 W/ LENET | $86.48_{\pm 0.22}$ |
| CIFAR-10 W/ RESNET-20 | $89.05_{\pm 0.44}$ |
| CIFAR-100 W/ LENET | $57.00_{\pm 0.09}$ |
| CIFAR-100 W/ RESNET-20 | $62.21_{\pm 0.85}$ |
| SHAKESPEARE | $52.00_{\pm 0.16}$ |
| STACKOVERFLOW | $28.50_{\pm 0.25}$ |
| GLDV2 | $74.03_{\pm 0.15}$ |

For this task, we have employed a two-layer Long Short-Term Memory (LSTM) classifier, consisting of 100 hidden units and an 8-dimensional embedding layer. Our objective is to predict the next character in a sequence, where there are a total of 80 possible character classes. The model takes in a sequence of 80

characters as input, and for each character, it learns an 8-dimensional representation. The final output of the model is a single character prediction for each training example, achieved through the use of 2 LSTM layers and a densely-connected layer followed by a softmax. This model architecture is the same used by (Li et al., 2020; Acar et al., 2021).

We report the result of centralized training for reference in Table 5: we train for 75 epochs with constant learning rate, using as hyperparameters 100 for the batch size, 1 for the learning rate, 0.0001 for the weight decay and no momentum.

**StackOverflow.** The Stack Overflow dataset is a language modeling corpus that comprises questions and answers from the popular Q&A website, StackOverflow. Initially, the dataset consists of 342477 unique users but for, practical reasons, we limit our analysis to a subset of $40k$ users. Our goal is to perform the next-word prediction on these text sequences. To achieve this, we utilize a Recurrent Neural Network (RNN) that first learns a 96-dimensional representation for each word in a sentence and then processes them through a single LSTM layer with a hidden dimension of 670. Finally, the model generates predictions using a densely connected softmax output layer. The model and the preprocessing steps are the same as in (Reddi et al., 2021). We report the result of centralized training for reference in Table 5: as per the hyperparameters, we use 16 for the batch size, $10^{-1/2}$ for the learning rate and no momentum or weight decay. We train for 50 epochs with a constant learning rate. Given the size of the test dataset, testing is conducted on a subset of them made by 10000 randomly chosen test examples, selected at the beginning of training.

**Large-scale Real-world Datasets.** As large-scale real-world datasets for our experimentation, we follow Hsu et al. (2020). GLDv2 is composed of $\approx 164k$ images belonging to $\approx 2000$ classes, realistically split among 1262 clients. INATURALIST is composed of $\approx 120k$ images belonging to $\approx 1200$ classes, split among 9275 clients. These datasets are challenging to train not only because of their inherent complexity (size of images, number of classes) but also because usually at each round a very small portion of clients is selected. In particular, for GLDv2 we sample 10 clients per round, while for INATURALIST we experiment with different participation rates, sampling 10, 50, or 100 clients per round. In the main paper, we choose to report the participation rate instead of the number of sampled clients to better highlight that the tested scenarios are closer to a cross-device setting, which is the most challenging for algorithms based on client participation, like SCAFFOLD and ours. As per the model, for both datasets, we use a MobileNetV2 pretrained on ImageNet.

**Details on the Experiment in Fig. 7.** In the main text (see Sec. 4.3) we provide an experiment to illustrate the convergence rate of GHBM (see Fig. 7). The learning problem consists in a linear regression of the coefficients $(a, b, c) \in \mathbb{R}$ of a quadratic function $f(x) = ax^2 + bx + c$. The synthetic dataset is made of 6400 observations of the above function (with $a = 10, b = 5, c = -1$) in the range $x \in [-10, 10]$. The dataset is split among $K = 50$ clients each one having 128 samples, and non-iidness is simulated by splitting the domain into equally big disjoint subsets, and having each client the observation of that domain.

Table 6: Details about datasets' split used for our experiments

|  | CIFAR-10 | CIFAR-100 | SHAKESPEARE | STACKOVERFLOW | GLDv2 | INATURALIST |
|---|---|---|---|---|---|---|
| Clients | 100 | 100 | 100 | 40.000 | 1262 | 9275 |
| Number of clients per round | 10 | 10 | 10 | 50 | 10 | $\{10, 50, 100\}$ |
| Number of classes | 10 | 100 | 80 | 10004 | 2028 | 1203 |
| Avg. examples per client | 500 | 500 | 2000 | 428 | 130 | 13 |
| Number of local steps | 8 | 8 | 20 | 27 | 13 | 2 |
| Average participation (round no.) | 1k | 1k | 25 | 1.5 | 40 | $\{5, 27, 54\}$ |

## C.2 Simulating Heterogeneity

For CIFAR-10/100 we simulate arbitrary heterogeneity by splitting the total datasets according to a Dirichlet distribution with concentration parameter $\alpha$, following Hsu et al. (2020). In practice, we draw a multinomial $q_i \sim \mathbf{Dir}(\alpha p)$ from a Dirichlet distribution, where $p$ describes a prior class distribution over $N$ classes, and $\alpha$ controls the heterogeneity among all clients: the greater $\alpha$ the more homogeneous the clients' data distributions will be. After drawing the class distributions $q_i$, for every client $i$, we sample training examples for each class according to $q_i$ without replacement.

Table 7: Hyper-parameter search grid for each combination of method and dataset (for $\alpha = 0$). The best values are indicated in **bold**.

| METHOD | HPARAM | CIFAR-10/100 | | SHAKESPEARE | STACKOVERFLOW |
|---|---|---|---|---|---|
| | | LENET | RESNET-20 | | |
| ALL FL | wd | [**0.001**, 0.0008, 0.0004] | [0.0001, **0.00001**] | [0, **0.0001**, 0.00001] | [**0**, 0.0001, 0.00001] |
| | $B$ | 64 | 64 | 100 | 16 |
| FEDAVG | $\eta$ | [2, **1.5**, 1, 0.5, 0.1] | [1.5, **1**, 0.1] | [1.5, **1**, 0.5, 0.1] | [1.5, **1**, 0.5, 0.1] |
| | $\eta_l$ | [0.1, **0.05**, 0.01, 0.005] | [1, **0.5**, 0.1, 0.01] | [1.5, **1**, 0.5, 0.1] | [1, 0.5, **0.3**, 0.1] |
| FEDPROX | $\eta$ | [2, **1.5**, 1, 0.5, 0.1] | [1.5, **1**, 0.1] | [1.5, **1**, 0.5, 0.1] | [1.5, **1**, 0.5, 0.1] |
| | $\eta_l$ | [0.1, 0.05, **0.01**, 0.005] | [1, **0.5**, 0.1, 0.01] | [1.5, **1**, 0.5, 0.1] | [1, 0.5, **0.3**, 0.1] |
| | $\mu$ | [1, 0.1, **0.01**, 0.001] | [1, **0.1**, 0.01, 0.001] | [0.1, 0.01, 0.001, **0.0001**, 0.00001] | [0.1, **0.01**, 0.001, 0.0001] |
| SCAFFOLD | $\eta$ | [1.5, **1**, 0.5, 0.1] | [1.5, **1**, 0.1] | [1.5, **1**, 0.5, 0.1] | [1.5, **1**, 0.5, 0.1] |
| | $\eta_l$ | [0.1, 0.05, **0.01**, 0.005] | [0.5, **0.1**, 0.01] | [1.5, **1**, 0.5, 0.1] | [1, 0.5, **0.3**, 0.1] |
| FEDDYN | $\eta$ | [1.5, **1**, 0.5, 0.1] | [1.5, **1**, 0.1] | [1.5, **1**, 0.5, 0.1] | [1.5, **1**, 0.5, 0.1] |
| | $\eta_l$ | [0.1, 0.05, **0.01**, 0.005] | [0.1, **0.01**, 0.005] | [1.5, **1**, 0.5, 0.1] | [1, 0.5, **0.3**, 0.1] |
| | $\alpha$ | [0.1, 0.01, **0.001**, 0.0001] | [0.1, 0.01, **0.001**, 0.0001] | [0.1, **0.009**, 0.001] | [**0.1**, 0.009, 0.001] |
| ADABEST | $\eta$ | [1.5, **1**, 0.5, 0.1] | [1.5, **1**, 0.5, 0.1] | [1.5, **1**, 0.5, 0.1] | [1.5, **1**, 0.5, 0.1] |
| | $\eta_l$ | [0.1, 0.05, **0.01**, 0.005] | [0.1, 0.05, **0.01**, 0.005] | [1.5, **1**, 0.5, 0.1] | [1, 0.5, **0.3**, 0.1] |
| | $\alpha$ | [0.1, 0.01, **0.001**, 0.0001] | [0.1, 0.01, **0.001**, 0.0001] | [0.1, **0.009**, 0.001] | [**0.1**, 0.009, 0.001] |
| MIME | $\eta$ | [2, **1.5**, 1, 0.5, 0.1] | [2, **1.5**, 1, 0.1] | [1.5, **1**, 0.5, 0.1] | [1.5, **1**, 0.5, 0.1] |
| | $\eta_l$ | [0.1, 0.05, **0.01**, 0.005] | [0.5, **0.1**, 0.01] | [1.5, **1**, 0.5, 0.1] | [1, 0.5, **0.3**, 0.1] |
| FEDAVGM | $\eta$ | [1, 0.5, 0.1, **0.05**, 0.01] | [1, **0.1**, 0.05] | [1, **0.5**, 0.1] | [1.5, **1**, 0.5, 0.1] |
| | $\eta_l$ | [0.5, **0.1**, 0.05, 0.01, 0.005] | [1, **0.5**, 0.1, 0.01] | [1.5, **1**, 0.5, 0.1] | [1, 0.5, **0.3**, 0.1] |
| | $\beta$ | [0.99, 0.9, **0.85**, 0.8] | [0.99, 0.9, **0.85**, 0.8] | [0.99, **0.9**, 0.85] | [0.99, **0.9**, 0.85] |
| FEDACG | $\eta$ | [1, 0.5, 0.1, **0.05**, 0.01] | [1, **0.1**, 0.05] | [0.5, **0.1**, 0.05] | [1.5, **1**, 0.5, 0.1] |
| | $\eta_l$ | [0.5, **0.1**, 0.05, 0.01, 0.005] | [0.5, **0.1**, 0.01] | [1.5, **1**, 0.5, 0.1] | [1, 0.5, **0.3**, 0.1] |
| | $\lambda$ | [0.99, **0.9**, 0.85] | [0.99, **0.9**, 0.85] | [0.99, **0.9**, 0.85] | [0.99, **0.9**, 0.85] |
| | $\beta$ | [0.1, **0.01**, 0.001] | [0.1, **0.01**, 0.001] | [0.1, 0.01, 0.001, **0.0001**, 0.00001] | [0.1, **0.01**, 0.001, 0.0001] |
| MIMEMOM | $\eta$ | [1, 0.5, **0.1**, 0.05] | [1.5, **1**, 0.5, 0.3, 0.1, 0.05] | [1, 0.5, **0.1**, 0.05] | [1.5, **1**, 0.5, 0.1] |
| | $\eta_l$ | [0.1, 0.05, **0.01**, 0.005] | [0.5, 0.1, 0.05, 0.03, **0.01**, 0.005] | [1.5, **1**, 0.5, 0.1] | [1, 0.5, 0.3, **0.1**, 0.05] |
| | $\beta$ | [0.99, 0.95, **0.9**, 0.85, 0.8] | [0.99, 0.95, 0.9, **0.85**, 0.8] | [0.99, **0.9**, 0.85] | [0.99, **0.9**, 0.85] |
| MIMELITEMOM | $\eta$ | [1, 0.5, **0.1**, 0.05] | [1.5, **1**, 0.5, 0.3, 0.1] | [1, 0.5, **0.1**, 0.05] | [1.5, **1**, 0.5, 0.1] |
| | $\eta_l$ | [0.1, 0.05, **0.01**, 0.005] | [0.1, 0.05, 0.03, **0.01**, 0.005] | [1.5, **1**, 0.5, 0.1] | [1, 0.5, 0.3, **0.1**, 0.05] |
| | $\beta$ | [0.99, **0.9**, 0.85, 0.8] | [0.99, 0.95, 0.9, **0.85**, 0.8] | [0.99, **0.9**, 0.85] | [0.99, **0.9**, 0.85] |
| FEDCM | $\eta$ | [1, 0.5, **0.1**, 0.05] | [1.5, **1**, 0.5, 0.1] | [1, 0.5, **0.1**, 0.05] | - |
| | $\eta_l$ | [1, 0.5, **0.1**, 0.05] | [1, 0.5, **0.1**, 0.5] | [1.5, **1**, 0.5, 0.1] | - |
| | $\alpha$ | [0.05, **0.1**, 0.5] | [0.05, **0.1**, 0.5] | [0.05, **0.1**, 0.5] | - |
| **GHBM (ours)** | $\eta$ | [**1**, 0.5, 0.1] | [**1**, 0.1] | [**1**, 0.5, 0.1] | [**1**, 0.5, 0.1] |
| | $\eta_l$ | [0.1, 0.05, **0.01**] | [0.1, **0.01**] | [**1**, 0.5, 0.1] | [1, 0.5, **0.3**, 0.1] |
| | $\beta$ | [**0.9**] | [**0.9**] | [**0.9**] | [**0.9**] |
| | $\tau$ | [5, **10**, 20, 40] | [5, **10**, 20, 40] | [5, **10**, 20, 40] | [5, 10, **20**, 40] |
| **FedHBM(ours)** | $\eta$ | [**1**, 0.5, 0.1] | [**1**, 0.1] | [**1**, 0.5, 0.1] | [**1**, 0.5, 0.1] |
| | $\eta_l$ | [0.1, 0.05, **0.01**] | [0.1, **0.01**] | [**1**, 0.5, 0.1] | [1, 0.5, **0.3**, 0.1] |
| | $\beta$ | [**1**, 0.99, 0.9] | [**1**, 0.99, 0.9] | [**1**, 0.99, 0.9] | [**1**, 0.99, 0.9] |

## C.3 Evaluating Communication and Computational Cost

In the main paper we showed a comparison in communication and computational cost of state-of-art FL algorithms compared to our solutions GHBM and FEDHBM: in this section we detail how those results in table Tab. 4 have been obtained. We follow a three-step procedure:

1. For each algorithm $a$, we calculate the minimum number of rounds $r_a$ to reach the performance of FEDAVG, the total amount of bytes exchanged $b_a$ in the whole training budget (number of rounds, as described in Appendix C.5) and the measure the corresponding total training time $t_a$. In this way, the different requirements in communication and computation of each algorithm are taken into account for the next steps.

2. We calculate the actual communication and computational requirements as $(tb_a = b_a \cdot s_a, tt_a = t_a \cdot s_a)$, where $s_a = \frac{r_a}{T}$ is the speedup of the algorithm w.r.t. FEDAVG. For those competitor algorithms that did not reach the target performance (*e.g.* MIMEMOM) in the training budget $T$, we conservatively consider $r_a = T$. In this way, the convergence speed of each algorithm is taken into account for determining the actual amount of computation needed.

3. We complement the above information with with a reduction/increase factor w.r.t. FEDAVG, calculated as $rtb_a = \left(1 - \frac{tb_a}{tb_{\text{FEDAVG}}}\right)$ and $rtt_a = \left(1 - \frac{tt_a}{tt_{\text{FEDAVG}}}\right)$ and expressed as a percentage. A cost reduction

(*i.e.* $rtb_a > 0$ or $rtt_a > 0$) is indicated with ↓, while a cost increase (*i.e.* $rtb_a < 0$ or $rtt_a < 0$) is indicated with ↑. This gives a practical indication of how much communication/computation have been saved in choosing the algorithm at hand as an alternative for FEDAVG.

## C.4 Hyperparameters

For ease of consultation, we report the hyper-parameters grids as well as the chosen values in Table 7. For GLDv2 and INATURALIST we only test the best SOTA algorithms: FEDAVG and FEDAVGM as baselines, SCAFFOLD and MIMEMOM.

**MobileNetV2.** For all algorithms we perform $E = 5$ local epochs, and searched $\eta \in \{0.1, 1\}$ and $\eta_l \in \{0.01, 0.1\}$, and found $\eta = 0.1, \eta_l = 0.1$ works best for FEDAVGM, while $\eta = 1, \eta_l = 0.1$ works best for the others. For INATURALIST, we had to enlarge the grid for SCAFFOLD and MIMEMOM: for both we searched $\eta \in \{10^{-3/2}, 10^{-1}, 10^{-1/2}, 1\}$ and $\eta_l \in \{10^{-2}, 10^{-3/2}, 10^{-1}, 10^{-1/2}\}$.

**ViT-B\16.** For all algorithms we perform $E = 5$ local epochs, and searched $\eta \in \{0.1, 1\}$ and $\eta_l \in \{0.03, 0.01\}$ following (Steiner et al., 2022), and found $\eta = 0.1, \eta_l = 0.03$ works best for FEDAVGM, while $\eta = 1, \eta_l = 0.03$ works best for the others.

## C.5 Implementation Details

We implemented all the tested algorithms and training procedures in a single codebase, using PYTORCH 1.10 framework, compiled with CUDA 10.2. The federated learning setup is simulated by using a single node equipped with 11 Intel(R) Core(TM) i7-6850K CPUs and 4 NVIDIA GeForce GTX 1070 GPUs. For the large-scale experiments we used the computing capabilities offered by LEONARDO cluster of CINECA-HPC, employing nodes equipped with 1 CPU Intel(R) Xeon 8358 32 core, 2,6 GHz CPUs and 4 NVIDIA A100 SXM6 64GB (VRAM) GPUs. The simulation always runs in a sequential manner (on a single GPU) the parallel client training and the following aggregation by the central server.

**Practicality of Experiments.** Under the above conditions, a single FEDAVG experiment on CIFAR-100 takes ≈ 02:05 hours (CNN, with $T = 20.000$) and ≈ 03:36 hours (RESNET-20, with $T = 10.000$). For SCAFFOLD we always use the `"option II"` of their algorithm (Karimireddy et al., 2020) to calculate the client controls, incurring almost no overhead in our simulations. We found that using `"option I"` usually degrades both final model quality and requires almost double the training time, due to the additional forward+backward passes. Conversely, all MIME's methods incur a significant overhead due to the additional round needed to calculate the full-batch gradients, taking ≈ 10:40 hours for CIFAR-100 with RESNET-20. On SHAKESPEARE and STACKOVERFLOW, FEDAVG takes ≈ 22 minutes and ≈ 3.5 hours to run respectively $T = 250$ and $T = 1500$ rounds.

## C.6 Additional Experiments

**Experiments on Cifar-10** Table 8 reports the results of experiments analogous to the ones presented in Tab. 2. For the main paper, we report experiments on CIFAR-100, as it is a more complex dataset and often a more reliable testing ground for FL algorithms. Indeed, sometimes algorithms perform well on CIFAR-10 but worse on CIFAR-100 (as for the already discussed case of FEDDYN). Results in Tab. 8 confirm the findings of the main paper: under extreme heterogeneity, some algorithms behave inconsistently across CNN and RESNET-20 (notice that FEDDYN and MIMELITEMOM only with CNN improve FEDAVG. Conversely, LO-CALGHBM and FEDHBM both consistently improve the state-of-art by a large margin.

Table 8: **Test accuracy (%) comparison of SOTA FL algorithms in a controlled setting.** Best result is in **bold**, second best is underlined.

| METHOD | CIFAR-10 (RESNET-20) | | CIFAR-10 (CNN) | |
|---|---|---|---|---|
| | NON-IID | IID | NON-IID | IID |
| FEDAVG | $61.0_{\pm1.0}$ | $86.4_{\pm0.2}$ | $66.1_{\pm0.3}$ | $83.1_{\pm0.3}$ |
| FEDPROX | $61.0_{\pm1.8}$ | $86.7_{\pm0.2}$ | $66.1_{\pm0.3}$ | $83.1_{\pm0.3}$ |
| SCAFFOLD | $71.8_{\pm1.7}$ | $86.8_{\pm0.3}$ | $74.8_{\pm0.2}$ | $82.9_{\pm0.2}$ |
| FEDDYN | $60.2_{\pm3.0}$ | $87.0_{\pm0.3}$ | $70.9_{\pm0.2}$ | $83.5_{\pm0.1}$ |
| ADABEST | $73.6_{\pm3.0}$ | $86.7_{\pm0.5}$ | $66.1_{\pm0.3}$ | $83.1_{\pm0.4}$ |
| MIME | $53.7_{\pm2.9}$ | $86.7_{\pm0.1}$ | $75.1_{\pm0.5}$ | $83.1_{\pm0.2}$ |
| FEDAVGM | $66.0_{\pm2.2}$ | $87.7_{\pm0.3}$ | $67.6_{\pm0.3}$ | $83.6_{\pm0.3}$ |
| FEDCM$_{(GHBM\ \tau=1)}$ | $65.2_{\pm3.2}$ | $87.1_{\pm0.3}$ | $69.0_{\pm0.3}$ | $83.4_{\pm0.3}$ |
| FEDADC$_{(GHBM\ \tau=1)}$ | $65.7_{\pm3.0}$ | $87.1_{\pm0.2}$ | $66.1_{\pm0.3}$ | $83.4_{\pm0.3}$ |
| MIMEMOM | $69.2_{\pm3.6}$ | $88.0_{\pm0.1}$ | $80.9_{\pm0.4}$ | $83.1_{\pm0.2}$ |
| MIMELITEMOM | $57.0_{\pm0.9}$ | $88.0_{\pm0.4}$ | $78.8_{\pm0.4}$ | $83.2_{\pm0.3}$ |
| **LocalGHBM (ours)** | $\underline{80.6}_{\pm0.3}$ | $\underline{88.8}_{\pm0.1}$ | $\underline{81.1}_{\pm0.3}$ | $\underline{83.7}_{\pm0.1}$ |
| **FedHBM (ours)** | $\mathbf{83.4}_{\pm0.3}$ | $\mathbf{89.2}_{\pm0.1}$ | $\mathbf{81.7}_{\pm0.1}$ | $\mathbf{83.8}_{\pm0.1}$ |

