# OpenReview forum: "Communication-Efficient Heterogeneous Federated Learning with Generalized Heavy-Ball Momentum"
_TMLR — Accepted by TMLR_

### Review · Reviewer_SJ4g · 2025-03-21

**Summary Of Contributions:**

The submission introduces a novel momentum-based method named GHBM for federated learning that addresses data heterogeneity and partial client participation. GHBM ensures convergence under unbounded heterogeneity, and its communication-efficient variant, FedHBM, matches FedAvg's communication cost while retaining momentum benefits. Theoretical guarantees and extensive experiments demonstrate GHBM's superior performance, faster convergence, and robustness compared to SOTA methods, offering new insights into momentum's role in federated learning.

**Audience:**

Yes

**Claims And Evidence:**

Yes

**Requested Changes:**

See weakness above.

**Strengths And Weaknesses:**

Strengths:
- The paper is overall very well written.
- The prior momentum-based federated learning paper (Cheng et al. 2024) proposed Fed-M and Scaffold-M, where the former only supports full client participation and the latter requires additional communication for control variates. This paper makes significant progress by effectively combining partial client participation and momentum without relying on control variates.
- The proposed method preserves the unbounded data heterogeneity property of momentum techniques.

Weaknesses:
- The partial participation scheme analyzed in the paper is limited to cyclic participation, which is an idealized scenario and may not reflect practical federated learning setups.
- The partial participation ratio $C$ should be explicitly defined. Additionally, the impact of $C$ and $\tau$ on the convergence results should be discussed in more depth. In its current form, Theorem 4.11 suggests that the convergence rate is independent of $C$ and $\tau$, which requires clarification or further justification.

---

> ### Author Response · Authors · 2025-04-03
> **Author Response 1/2**
>
> We sincerely thank the reviewer for their time and thoughtful feedback. We are pleased that they found our paper well-written and recognized its contributions to the FL research field. Below, we address the raised concerns and outline the corresponding revisions to our manuscript (indicated in violet for visual clarity).
>
> **W1: On our use of cyclic participation scheme**
> We thank the reviewer for bringing this concern to our attention, as this allows us to clarify why we use cyclic participation.
>
> The theoretical reason behind cyclic participation is related to the ability to determine the clients' contribution inside the average of the last $\tau$ pseudo-gradients since the important point behind the proof of GHBM is making sure that this $\tau$-averaged pseudo-gradient used for updating the momentum is not affected by heterogeneity (i.e. we need the quantity in lemma 4.7 to not depend on bounded heterogeneity). To verify this condition it is necessary to properly choose $\tau$. However, under random uniform, due to the non-zero probability of sampling the same client within $\tau$ rounds,  this condition is hardly verified.
> It is still technically possible to modify the algorithm such that happens even without cyclic sampling: we could explicitly keep track of the pseudo-gradients of each client, maintaining a uniform average of the most recent pseudo-gradients of each client, instead of processing aggregated pseudo-gradients (see discussion in footnote 2 of page 8). However, such an approach would be unrealistic to implement: for example, it would not be compliant with protocols like Secure Aggregation, widely used in real-world implementations, thus posing a significant practical limitation.
>
> In this light, instead of complicating the algorithm just for the sake of proving stronger theoretical results (i.e. for any sampling strategy), we preferred adopting cyclic participation and relying on extensive experiments to show that the advantages of GHBM also hold under random uniform client sampling, which is common in the literature.
>
> Let us also note that, even though purely cyclic participation may be an idealized setting, in practice it has been shown that similar periodic patterns emerge when client availability is correlated with geographical location (e.g. mobile phones charging at night) [1,2]. However, **cyclic participation is only used in theoretical analysis, never in our experiments** (see remark 4.6). In particular, all the experiments we present in section 5 are conducted under common uniform random client sampling.
>
> Regarding the concern on practical FL setups, we would also like to remark that our designed experimental settings accurately reflect real-world conditions, as in section 5.3 we report results using large-scale real-world federated datasets, with a large number of clients (on the order of $10^4$ -$10^5$) and considering very low client participation we may expect in real-world systems (e.g. $C < 1\\%$).
>
> In our opinion, maintaining an algorithm practical, simple, and flexible is a much more impactful strategy to advance the current state of FL.  Indeed, while several impactful works provide theoretically performant methods, their empirical evaluation and their real-world applicability are often limited.
>
> **Action on the manuscript:** We integrated the above clarifications into section A.2, which already discussed the theoretical and practical implications of cyclic participation for this work. We clarify that cyclic participation is a technical tool used in the analysis to show the theoretical advantages of GHBM and that cyclic training does not need to be imposed in practice for having good performance, as we show via extensive experimentation under uniform random client sampling.
>
> [1] Zhu et al., Diurnal or nocturnal? federated learning of multi-branch networks from periodically shifting distributions\
> [2] Crawshaw et al., Federated learning under periodic client participation and heterogeneous data: A new communication-efficient algorithm and analysis

---

> ### Author Response · Authors · 2025-04-03
> **Author Response 2/2**
>
> **W2: On the impact of $C$ and $\tau$ in the convergence rate**
>
> The participation ratio $C$ is the portion of clients participating in each round, i.e.,  the ratio between the number of selected clients per round and the total number of clients (which is denoted as $K$). While we chose not to assign a specific name to the number of selected clients per round, simply expressing it as $|S^t|=K\cdot C$, we acknowledge the suggestion and will add an explicit definition for $C$.
>
> Regarding the convergence rate, while $\tau$ and $C$ do not explicitly appear in the final bound, they both impact the convergence rate in the choice of the client learning rate $\eta_l$ in eq. (120). In fact, Thm. 4.11 requires $\eta_l \leq \mathcal{O}(\frac{1}{\sqrt{\tau}})$, and to guarantee convergence under unbounded heterogeneity we require $\tau=\frac{k}{C},\, \forall k \in \mathbb{N}^+$. Technically, a higher $\tau$ imposes a stricter bound on the client learning rate, so one would choose the minimum that allows convergence without bounded gradient heterogeneity, which is indeed $\tau=\frac{1}{C}$, as set in Thm. 4.11. As such, the convergence rate does depend on $\tau$ (and indirectly on $C$), as is expected.
>
> **Action on the manuscript:** We will clarify the dependence of the result in Thm. 4.11 on $\tau$ and $C$ right after the theorem statement in section 4.3 and add an explicit reference to where $\tau$ appears in the convergence rate.
>
> We hope our answer clarifies the reviewer's concerns and we remain available for further discussion.

---

> > ### Comment · Reviewer_SJ4g · 2025-04-17
> >
> > I thank the authors for their response. It remains unclear to me how the final convergence bound in Thm. 4.11 relies on the parameters $\tau$ and $C$. In the proof, the approximate inequalities (125)-(128) seem to hide the dependence on these constants. I think this dependence should be made explicit both in the proof and in the statement of the main theorem.

---

> > > ### Author Response · Authors · 2025-04-17
> > > **Further clarifications**
> > >
> > > We thank the reviewer for the feedback. In the following, we clarify the mentioned passage of the proof to explicitly show the dependence on $\tau$ and $C$.
> > >
> > > In eq. (126) we applied the upper bound on $\eta_l J L \leq \sqrt{\frac{L \Delta}{\beta^3 \tau G_\tau T}}$ as imposed in eq. (120). Let us provide the detailed passage on the third term in the RHS of eq. (126):
> > > \\[
> > > (\eta_l J L \beta)^2 \tau G_\tau \leq \beta^2 \left(\frac{L \Delta}{\beta^3 \tau G_\tau T}\right) \tau G_\tau = \frac{L \Delta}{\beta T}
> > > \\]
> > >
> > > Hence, this passage leads from eq. (126) to:
> > >
> > > $
> > > \\begin{aligned}
> > > LHS &\lesssim \frac{L\Delta}{T} + \frac{2L\Delta}{\beta T} +  \frac{L\Delta}{\beta T} + \beta \frac{\sigma^2}{|\mathcal{S}|J} \\\\
> > > &=  \frac{L\Delta}{T} + \frac{3L\Delta}{\beta T} + \beta \frac{\sigma^2}{|\mathcal{S}|J} \\\\
> > > &\lesssim \frac{L\Delta}{T} + \frac{L\Delta}{\beta T} + \beta \frac{\sigma^2}{|\mathcal{S}|J}
> > > \\end{aligned}
> > > $
> > >
> > > which was the result in eq. (127).
> > >
> > > **We followed the reviewer’s suggestion in the latest revision and made the dependence more explicit, both in the proof**, where we added the above additional passage showing explicit dependence from $\tau$, **and in the statement of Thm. 4.11**, explicitly stating that $\eta_l \leq \mathcal{O}(\frac{1}{\sqrt{\tau}})$ and highlighting the constraint $\tau=\frac{1}{C}$ in the discussion right below.
> > >
> > > We hope these additional clarifications dispel the doubt on the matter.

---

### Review · Reviewer_5YNc · 2025-03-25

**Summary Of Contributions:**

This paper addresses federated optimization in cross-silo (finite-sum) settings, particularly focusing on scenarios with a large number of clients where methods relying on control variates (such as SCAFFOLD) struggle or are impractical. To this end, the authors propose a novel Generalized Heavy-Ball Momentum (GHBM) approach, which, when combined with local steps, results in a new federated optimization method. The paper provides a theoretical analysis of GHBM under a cyclic partial participation assumption, demonstrating its convergence irrespective of data heterogeneity. The work then introduces practical modifications to GHBM, along with extensive experimental results on several federated learning benchmarks, highlighting the benefits of the proposed approach.

**Audience:**

Yes

**Broader Impact Concerns:**

No concerns for this kind of work

**Claims And Evidence:**

Yes

**Requested Changes:**

### Critical Changes

- The hyperparameter tuning in Table 8 appears insufficient, as many best-performing values are at the edge of the search grid. A more thorough tuning process or justification for the current choices is needed to ensure fair comparisons.

- The choice of measuring "final model quality" over the last 100 rounds of training is questionable. Full convergence curves should be provided to better illustrate the training dynamics and justify this metric.

- The convergence rate table (Table 1) does not include a heterogeneity bound for GHBM, which might create the misleading impression that it does not provide convergence advantages. This should be addressed.

- The term "uniform sampling" needs to be explicitly defined. Does it refer to independent/Poisson sampling of clients, or mini-batch sampling with or without replacement?

- The condition on $\eta_l$ in Equation (120) requires decreasing the local step size proportionally to the number of local steps. This hinders the benefits of local updates compared to methods like ProxSkip/SPPM. The authors should clarify how Theorem 4.11 compares to standard distributed (mini-batch) SGD.

- The concept of cyclic participation should be explained in more detail. Are schemes like Random Reshuffling or Shuffle-once considered instances of cyclic participation? The assumptions and implications need clarification.

### Recommended Improvements

- Figure 1 appears to be unreferenced in the text. Additionally, it lacks details about the experimental setup, such as which method is used and what the heterogeneity notation $k$ refers to.

- The experiments presented in Figure 2 lack setup details, making it difficult to interpret the results. A clear description should be added. Similarly, for Table 8, explicit information on the tuning process should be provided.

- The meaning of "compressed updates" after the reference to (Das et al., 2022) on page 3 is unclear. The authors should specify what type of compression is being referred to.

- A brief review of the benefits of momentum in the centralized setting would be useful for context.

- The inequality in Corollary 4.8 appears to be missing an expectation on the left-hand side. This should be checked and corrected if necessary.

- The term $g^{t_\tau}$ in Lemma 4.10 is not defined. It should be explicitly introduced to improve readability.

- The paper should clarify whether the communication cost model considers only client-to-server communication or a bidirectional model.

- The authors should discuss the compatibility of the proposed methods with FedOpt and adaptive local optimizers.

- The statement that the work "has not been published in peer-reviewed venues" seems unnecessary. Many influential federated learning papers remain unpublished on arXiv. The authors should either remove this comment or clarify its intent.

**Strengths And Weaknesses:**

## Strengths

- The proposed generalization of Heavy-Ball Momentum (GHBM) introduces a novel and interesting approach with potential applications beyond federated learning. The convergence of GHBM is analyzed under relatively mild assumptions, with the exception of the cyclic participation assumption.

- The paper is well-structured and clearly presented, making it easier to understand the proposed method and its theoretical foundations.

- The experimental study is extensive, covering a range of practical federated learning benchmarks and providing strong empirical evidence for the method’s efficacy. This is a significant strength compared to many purely theoretical contributions in the field.

## Weaknesses

1. Table 8 indicates that, for many method-problem pairs, several hyperparameters appear to be "undertuned," as the highlighted (bold) best-performing values are often at the boundary of the search grid. This raises concerns about the fairness of comparisons and the reliability of the reported results.

2. The choice of "final model quality," measured over the last 100 rounds of training, is questionable. Full convergence curves would provide a more comprehensive understanding of training dynamics and ensure that the chosen metric accurately represents overall performance.

3. The approach to controlling heterogeneity using a parametrized Dirichlet distribution might limit the generalization of results, as it accounts only for label heterogeneity. This makes results for different values of $\alpha$ difficult to compare meaningfully.

4. > "The BGD assumption (4.3) is not necessary, as the two terms on the left-hand side (LHS) of the above inequality are the same by definition."

This sentence is ambiguous. Does this imply that one can show the right-hand side (RHS) of the inequality is effectively zero without relying on the BGD assumption, or is the assumption still required for the proof?

 More generally, illustrating this effect from a purely distributed SGD perspective would be insightful. If I understand correctly, results such as Lemma 4.7 stem primarily from the sampling strategy rather than the use of momentum. Is this interpretation correct?

5. The condition on $\eta_l$ in Equation (120) requires the local step size to decrease proportionally to the number of local steps. This hinders the benefits of performing local updates, unlike methods such as ProxSkip or SPPM. How does Theorem 4.11 compare to standard distributed (mini-batch) SGD in this regard?

### Minor

- The discussion of related work, particularly Mishchenko et al. (2022), lacks sufficient detail. Since their method is stateless and supports partial participation, further clarification of the differences would help contextualize the contribution.

- The presentation could be improved by providing a simplified explanation or illustration of core concepts, such as cyclic participation.

- There is an incorrect reference to Karimireddy et al. (2020) at the beginning of page 2.

---

> ### Author Response · Authors · 2025-04-03
> **Author Response 1/4**
>
> We sincerely thank the reviewer for their time and thoughtful feedback. We are pleased that they appreciated the novelty of our proposed approach and its potential influence beyond federated learning. We are also particularly pleased that the extent of our experimental study has been recognized as a factor of merit of this submission, compared to many other works in the area that are purely theoretical. Below, we address the raised concerns and outline the corresponding revisions to our manuscript (indicated in violet for visual clarity).
>
> **W1: About hyperparameter tuning**
> Thanks for expressing your concern as it allows us to specify more details on our tuning process. The hyperparameters whose optimal value is often at the edge of our grid are mainly the learning rates. To tune them, we followed the guidelines reported in prior experimental works [5], sweeping $\eta_{eff}=\eta\eta_l$ from $1$ to $10^{-3}$ and in general agree with their finding that usually $\eta_{eff}=10^{-2}$ works best. To the best of our knowledge, values of $\eta>1$ and $\eta_l<10^{-2}$ are not used in previous FL works, which is why we did not expand in this direction. However, please note that we did expand the grid (only for competitors), making it more fine-grained, when the performance of competitors was unsatisfactory, e.g. much lower than expected, especially in large-scale scenarios, to make sure our findings are reliable. For other hyperparameters, we followed the indications in respective works. **In all cases, we tuned the hyperparameters of competitors more than we did for our methods, remaining fair in that sense.**
>
> To further demonstrate our efforts in guaranteeing fair evaluations of competitors, we ran an additional search for all competitor methods (not for ours) for hyperparameters that were off the grid initially reported in our submission, ensuring the best-performing values are within the new grid. As we report in our revision, sometimes we have slightly better results ($+2\\%,+3\\%$ w.r.t the values we previously obtained). Nevertheless, **the gap with our methods (which we did not tune any further) is still $>12\\%$ and the full convergence curve shows starkly faster convergence (figure 7), so our findings are confirmed.**
>
> Since we consider a lot of competitors and the grid search is already very comprehensive, at the time of this rebuttal we limited ourselves to the combination ResNet-20/CIFAR-100 (which in our experience is the most challenging). Experiments on all the other combinations model/dataset/competitor are currently running and will be added to the final version.
>
> **W2: Clarification on final model quality metric and full convergence curves**
> In our paper, we decided to look at the performance of the trained model and its convergence speed separately, to enhance clarity and ease of comparison. This is typically more difficult using convergence curves when many competitor algorithms are included in the study, as in our case. In particular, Tables 2-3 evaluate the results at convergence, while Table 4 evaluates the speed of the algorithms, using directly the communication cost (total number of bytes sent over the network) and computation cost (wall-clock time of simulation) to reach the same result as FedAvg.
> Regarding the final model quality metric, we use it to show the result at convergence and consider the average accuracy of the last 100 rounds to ensure a fair and reliable measurement over using just the last round. We also highlight that all experiment results are averaged across 5 random seeds.
>
> Having clarified that our choices come from an effort to present results in the clearest way, **in our revision we include full convergence curves as requested (see Fig 7).**

---

> ### Author Response · Authors · 2025-04-03
> **Author Response 2/4**
>
> **W3: Simulating heterogeneity:**
> We acknowledge that the approach we used to simulate heterogeneity in academic datasets (e.g. CIFAR-10/100) using a parametrized Dirichlet distribution only accounts for label heterogeneity. However, it must be noted that simulating label heterogeneity is the common approach to simulating statistical heterogeneity in FL.
>
> In particular **the choice of using a parametrized Dirichlet distribution is a common practice** in previous FL works in the area of this submission. For example, it is used by FedDyn [1], FedCM [2,3], and other papers evaluating FL algorithms [4,5]. Each value of $\alpha$ should be seen as a “severity level” of simulated heterogeneity, so it is correct that experiments with splits generated with different values of $\alpha$ should not be compared to each other.
> Even though other forms of heterogeneity can be of interest to consider, our simulation protocol adheres to the common practice in literature for the approaches in the same area as GHBM.
>
> Please also notice that, besides simulating heterogeneity in academic datasets, in our work we consider realistic cases using real-world FL datasets, as we are pleased the reviewer appreciated it as one of the strengths of our work. In those datasets the heterogeneity arises from real-world data distribution - thus considering _real_ heterogeneity (we refer to [5] for full details on Landmarks and INaturalist datasets), not just the one we can simulate on smaller scale datasets.
>
> [1] Acar et al., Federated learning based on dynamic regularization, ICLR 2021\
> [2] Xu et al., FedCM: Federated learning with client-level momentum, 2021.\
> [3] Cheng et al., Momentum benefits non-iid federated learning simply and provably, ICLR 2024\
> [4] Hsu et al., Measuring the effects of non-identical data distribution for federated visual classification, 2019\
> [5] Hsu et al., Federated visual classification with real-world data distribution, ECCV 2021
>
> **W4: Clarification about remark 4.9**
> The remark serves to point out that, under assumption 4.4 and the constraint $\tau=1/C$, **the RHS of the inequality is effectively zero without the need of using the BGD assumption**, because the difference in the LHS is trivially zero by definition (i.e. the average gradient over all clients data is, by definition, a full gradient). Without that remark, a reader may mistakenly think that we claim the RHS is zero because $(1- \tau \cdot C)$ is equal to zero for $\tau=1/C$, which would not be valid because unbounded heterogeneity means that $G \rightarrow \infty$.
> The point of showing these intermediate lemmas in the main text is showing the effect of $\tau$ on heterogeneity reduction, building intuition on why for the value of $\tau$ imposed by Thm 4.11 ($\tau=1/C$) heterogeneity is completely removed. Indeed, the proof of Thm. 4.11 does not use the BGD assumption.
>
> Let us also clarify that Lemma 4.7 bounds the effect of heterogeneity in the gradient used in the update of momentum, in particular showing that if we do not have access to the information of all clients, residual heterogeneity remains.
>
> As such, **results presented primarily stem from the fact that our method uses a pseudo-gradient over all the clients to update the momentum** (i.e. $\tau$ is not equal to one as in classical momentum, but it is set such that we completely remove heterogeneity, as discussed above). Cyclic participation assumption is needed just to know which client gradients contribute to the average of the server pseudo-gradients, and consequently derive the value of $\tau$ which realizes the condition of having a uniform average of all clients’ gradients. This is the passage from lemma 4.7, which does not assume any particular sampling strategy, to corollary 4.8.
>
> The final convergence result stems from using a momentum term unaffected by heterogeneity, and to have this condition in partial participation we need to access old gradients. So, in practice, we remark that the results **do NOT stem primarily from the sampling strategy**. This discussion was reported in section A.2 in our original submission, we will make sure to add additional pointers to it.
>
> Please let us know if these clarifications dispel the doubt or if additional discussion is needed.

---

> ### Author Response · Authors · 2025-04-03
> **Author Response 3/4**
>
> **W5: About convergence rate and local steps**
> As the reviewer correctly notes, our analysis does not show a benefit in the convergence rate from the local steps $J$, as Proxskip does. In this regard, our proof shares the same result as the currently best-known theory for momentum-based FL methods [3].
>
> However, GHBM does have a theoretical advantage related to local steps w.r.t standard distributed (mini-batch) SGD, which is clearer when adding local steps to incremental gradient methods. They refer to a family of algorithms in the context of finite-sum optimization of $n$ functions (similar to the FL setting we are in, where functions are clients) designed to match the performance of GD without re-evaluating all the $n$ functions at each iteration, which is analogous to overcoming the heterogeneity induced by partial participation in FL.
>
> One algorithm of this family, the Incremental Aggregated Gradient (IAG), has been proved to remove the effect of heterogeneity among functions, by approximating the full gradient with an aggregate of old gradients of each function, under the same notion of cyclic participation we use in this work [7]. However this holds in standard distributed mini-batch optimization, i.e. $J=1$.
>
> GHBM is based on a similar intuition of the IAG algorithm, but one key difference is that our approach applies this logic to the update of the momentum state, not to the gradient estimate. This is crucial when local steps are introduced, since simply extending IAG by adding local steps would not allow discharging the bounded heterogeneity assumption due to client drift as GHBM does. In fact, the convergence rate of IAG would be upper bounded by that of FedAvg in full participation, whose lower bound is known to be affected by heterogeneity (see theorem II of [6]). These additional connections with incremental gradient methods are provided in section A.3 of the manuscript.
>
> We hope this additional discussion clarifies that, while we do not directly prove the advantage of local steps (which is currently an open problem for momentum-based methods), GHBM takes into account the challenges implied by local steps and heterogeneity (i.e. the client drift), making a step forward in settings with partial client participation.
>
>
> [6] Karimireddy et al., SCAFFOLD: Stochastic Controlled Averaging for Federated Learning, ICML 2020\
> [7] Gürbüzbalaban et al., Convergence rate of incremental gradient and newton
> methods, SIAM Journal on Optimization
>
> **Minor weaknesses**\
> **M1:** The algorithm proposed by Mishchenko et al. (2022) can be seen as a combination of FedProx with SCAFFOLD/Scaffnew, and similarly relies on additional server control variates to correct the drift, so the underlying principle is still variance reduction (although with proper innovations authors explain in the paper). Quite differently, GHBM shows that a careful choice of momentum alone can tackle heterogeneity and partial participation in FL. We added this discussion to our revision.\
> **M2:** We added an illustration of cyclic participation in figure 5.\
> **M3:** Thanks for spotting the typo, we fixed the citation.
>
>
> **Response to requested changes**\
> **C2.** We added full convergence curves supplementing the results already presented in Tables (see fig 7).\
> **C3.** The convergence rate in Table 1 does not include an heterogeneity bound for GHBM because, under the assumptions of theorem 4.11, GHBM converges under unbounded heterogeneity (i.e. BGD is not needed, similarly to how we reported for FedCM in full participation from [3]). This is the exact reason why GHBM provides convergence advantages regarding heterogeneity.\
> **C4.** Random uniform client sampling means that all clients have the same probability to be sampled, and that at each round a subset of clients is selected independently without replacement. This is common practice in FL. We added this clarification in our revision.\
> **C6.** We added an illustration to visualize intuitively cyclic participation in figure 5, while implications of cyclic participation are discussed in section A.2 Random Reshuffling is not considered an instance of cyclic participation, since we need clients to be sampled at regular and equal intervals during the whole training. This means that, once a sampling order of clients has been fixed, it should remain the same. In this light, Shuffle-Once can be considered cyclic for assumption 4.4., meaning that at the beginning of training we can fix any sampling order and follow it cyclically during the whole training (i.e. do not reshuffle clients).

---

> ### Author Response · Authors · 2025-04-03
> **Author Response 4/4**
>
> **Response to suggested improvements**\
> **I1.** Figure 1 is referenced in section 3.3, right after the equations (4-5). The method used is FedAvg, but we run extra analysis on the pseudo-gradients to measure the distance between a $\tau$-averaged pseudo gradient and the pseudo gradient we would have in full participation. In detail, at each round:
> 1. we calculate the pseudo-gradients $\tilde{g}^t$ as per FedAvg rule, while keeping the statistics on $\tilde{g}^{t}_\tau$ (i.e. average pseudo-gradients of past $\tau$ rounds).
> 2. we calculate the pseudo-gradients over all the clients, at current parameters, and compute the difference with the previous term (this is the quantity shown in figure 1).
> 3. we update the model with $\tilde{g}^t$, exactly as FedAvg does.
> In practice the extra step w.r.t. FedAvg is (2), which does not change the updates of the algorithm. Please note that the code we will release allows full reproducibility of _all_ the results presented in the paper, including this analysis.
> The purpose of the figure is showing that increasing $\tau$ is effective in reducing the l2-distance w.r.t. a gradient over all the clients, even if those $\tau$ pseudo-gradients are calculated at old parameters. $\alpha=10k$ simply means $\alpha=10.000$ (i.e. “k” is a shorthand for $\times 1000$). We changed it to $\alpha=10.000$ for clarity.
>
> **I2.** We added full details in the appendix for the synthetic experiment in Figure 2. Along with the code, we will release the generated data for that experiment.\
> **I3.** (Das et al. 2022) uses quantization as a compression technique. We now added this detail in the text.\
> **I4.** To the best of our knowledge, momentum has no theoretical advantage over vanilla SGD in the non-convex centralized setting, even though the experimental advantages are widely recognized. We added a brief discussion in the appendix.\
> **I5.** Thanks for the note, we fixed the missing expectation.\
> **I6.** The term $g^t_\tau$ is previously defined in the statement of lemma 4.7, but we acknowledge that repeating it also in lemma 4.10 can improve readability, thanks for the suggestion.\
> **I7.** When evaluating the communication cost we consider the total number of bytes sent over the network, thus including both the downlink (server-to-clients) and uplink (clients-to-server) communication. We added clarification in section 5.3\
> **I8.** Our approach has a fundamental difference with the adaptive optimizers following the FedOpt framework. In FedOpt, the client optimizer is vanilla SGD, and an adaptive optimizer is on the server side (see their algorithm 2), while ours uses server-level statistics during local optimization. The necessity of using server-level statistics when the aim is to correct client drift is documented in several works, most notably the Mime paper, which regarding this aspect is more similar to ours. We can better state that our proposed method is compatible with Mime, more than with FedOpt. The core idea behind GHBM can be further extended to adaptive optimizers (e.g. Adam), and potentially allow even better performance. As this introduces significant non-trivialities, we are actively pursuing this direction for future work.\
> **I9.** Peer-reviewed papers are proofread by reviewers, usually ensuring higher quality. However there is really no particular intent behind specifying it, we just removed that line as suggested.

---

> ### Author Response · Authors · 2025-04-08
> **Update on W1**
>
> Dear Reviewer, **we have now completed the extended grid search for ALL the method-problem pairs**, as we committed in our last message.
> With this extended hyperparameter tuning, we have now ensured that all the best hyperparameters fall within the edges of a very fine-grained grid (see Table 8 in the **latest revision**).
>
> We confirm that the benefit of such extreme tuning of competitor methods usually results in $+2\\%, +3\\%$ w.r.t the values we previously obtained.
> **Nevertheless, these improvements do not affect our findings, because the gap between the best competitors and our methods is unchanged and quite significant.** Please note that we did this extended tuning only for the competitors and not for our method, to provide further evidence that our methods do not need such an extensive hyperparameter search to largely outperform the competitors, even when they are extremely tuned.
>
> We hope that this resolves the initial concern.
> As the rebuttal period is approaching its end, please let us know if any additional discussion is needed, we are available for discussion.

---

> > ### Comment · Reviewer_5YNc · 2025-04-09
> > **Response to rebuttal**
> >
> > I would like to thank the authors for their detailed responses, which provided insightful discussions and clarifications. I also appreciate the incorporation of the suggested revisions into the manuscript. At this point, my major concerns have been adequately addressed.
> >
> > I would particularly like to highlight the following responses as especially insightful:
> >
> > - **W5:** The intuition provided around Incremental Aggregated Gradient (IAG) was very helpful. I recommend including this explanation in the main text, as it significantly improves understanding of the idea.
> >
> > - **C6:** The visualization of cyclic participation patterns, although more constrained than the scenarios I initially envisioned, is still very informative and a valuable addition to the paper.
> >
> > Regarding **I8**, I would appreciate further elaboration on why extending the GHBM idea to adaptive optimizers like Adam introduces "significant non-trivialities." Adaptive methods with server-side updates (e.g., FedAdam) are often among the best-performing approaches in realistic federated learning scenarios, so clarifying the challenges involved would be useful.
> >
> > Finally, I believe that the discussion comparing this work to that of Cho et al. (2023), currently located in the Appendix, is important for situating the contribution within the broader literature. It would be better placed in the main body of the paper to ensure it receives appropriate visibility.

---

> > > ### Author Response · Authors · 2025-04-10
> > > **Further elaboration on I8**
> > >
> > > We thank the reviewer for the quick response and useful feedback, we are pleased our responses addressed the concerns.
> > > We agree that incorporating into the main text the discussion about Incremental Aggregated Gradient (IAG) and the contextualization with the work of Cho et al. (2023), which are currently in the Appendix, is a valuable enhancement to the paper. We followed this suggestion in our last revision.
> > >
> > > **I8: why extending the GHBM idea to adaptive optimizers like Adam introduces significant non-trivialities**\
> > > Thanks for the interest in further extending GHBM to adaptive optimizers. We agree that, given the success of adaptive methods, this is an interesting future direction.
> > > However, to truly translate the empirical success of Adam into FL, one should understand the reason underlying its improved performance.
> > >
> > > How exactly Adam works is still undecided - some works suggest Adam perform normalization [8], whereas others suggest it adapts to the changing landscape. Depending on which phenomenon underlies Adam's success, its federated counterpart will be different. To simulate the normalizing effect, the second order statistics should be independently calculated for each client. To adapt to changing smoothness, we would need to use global second-order statistics. Further, implementing adaptivity into local updates requires more communication. Thus, mitigating these costs and investigating various explanations for Adam's success and translating them into FL makes for an exciting future work.
> > >
> > > [8] Kunstner et al., Noise Is Not The Main Factor Behind The Gap Between Sgd And Adam On Transformers, But Sign Descent Might Be, ICLR 2023

---

### Review · Reviewer_1iVb · 2025-03-26

**Summary Of Contributions:**

This paper proposes Generalized Heavy-Ball Momentum (GHBM), a variant of momentum designed for Federated Learning (FL) to mitigate the effects of statistical heterogeneity and partial client participation. The key idea is to compute the momentum term by averaging past updates over multiple rounds, rather than relying solely on the most recent gradient information. The authors provide a theoretical analysis demonstrating that GHBM converges under arbitrary heterogeneity, particularly in cyclic partial participation settings. Additionally, they introduce FedHBM, a communication-efficient variant that retains the momentum benefits while maintaining the same communication complexity as FedAvg. Experimental results suggest that GHBM and FedHBM outperform existing FL algorithms, particularly in scenarios with extreme non-IID data and low client participation.

**Audience:**

No

**Broader Impact Concerns:**

The paper does not raise significant ethical concerns.

**Claims And Evidence:**

No

**Requested Changes:**

1. The paper should compare against more recent methods, especially those that target both momentum and communication efficiency, such as Communication-Efficient Federated Learning with Accelerated Client Gradient [1], to ensure a fair baseline comparison.

2. The authors should clarify that FedHBM’s communication advantage is only relative to FedCM (client momentum) and does not provide meaningful gains over FedAvgM (server momentum). The authors should also discuss the trade-off of requiring clients to store past model states, which could be a practical limitation in real FL deployments.

**Strengths And Weaknesses:**

**Strengths:**

1. The introduction of GHBM is conceptually straightforward and builds upon existing momentum-based methods, making it relatively easy to implement.

2. Theoretical analysis provides some insights into why classical momentum fails under partial participation and how GHBM mitigates this issue.

3. Extensive experiments are conducted on vision (CIFAR-10/100, GLDv2, INaturalist) and NLP (Shakespeare, StackOverflow) tasks, covering both controlled and real-world FL scenarios.


**Weaknesses:**

1. The core idea of GHBM is overly simplistic—essentially just applying a moving average over multiple past gradients, which is a well-known technique in optimization. It is unclear whether this warrants a standalone contribution.

2. The paper lacks comparisons with recent advances, such as [1], which also focus on momentum and communication efficiency in federated learning.

3. The claim that FedHBM is communication-efficient is questionable:

   - Compared to FedAvgM (server momentum), FedHBM does not provide significant communication savings, as FedAvgM already avoids sending extra momentum information.

   - FedHBM requires clients to store past model states, which may be impractical in scenarios where client devices have limited memory (e.g., mobile or edge devices).


[1] Communication-Efficient Federated Learning with Accelerated Client Gradient. CVPR 2024

---

> ### Author Response · Authors · 2025-04-03
> **Author Response 1/3**
>
> We thank the reviewer for their time and feedback. Below, we address the raised concerns and outline the corresponding revisions to our manuscript (indicated in violet for visual clarity).
>
> **W1: On the contribution value of this submission**
> On the claim that the simplicity of the core idea behind GHBM could undermine the contribution we bring with this submission, we must be clear in expressing our respectful disagreement. There are at least three aspects in which this submission represents a significant contribution:
>
> **Algorithmic design.**
> The design of our GHBM is built from careful consideration of the theoretical challenges posed by heterogeneity in partial participation: indeed, we start the paper by describing and demonstrating the issues of classical momentum in FL (i.e. FedCM) when dealing with heterogeneity and partial participation, and then step-by-step construct the GHBM formulation based on what is needed to overcome them. In particular, we show the modification we introduce has a specific theoretical rationale behind it,  as **we demonstrate both in theory and in practice why the (classical) momentum update rule should be modified as we propose for obtaining our strong results**.
> Moreover, GHBM is not the only algorithmic contribution of this paper: there are also communication-efficient variants, which perform similarly to GHBM and avoid exchanging additional data. We also discuss the applicability of GHBM and variants based on the FL scenario at hand, highlighting when one algorithm should be preferred over another (see second paragraph of section 3.4).
>
> As long as a proposed method is grounded and proven to be effective, which clearly is the case of this submission, algorithmic simplicity is a further advantage, as we are happy to see it mentioned among the strengths of our approach in the review.
>
> **Theoretical results.**
> In our work we prove that the proposed GHBM converges under arbitrary heterogeneity in (cyclic) partial participation, advancing over prior work, which proves this result for FedCM in full participation only. **Removing the restriction of full participation, is indeed a significant result**, which offers new insights into momentum's role in federated learning.
>
> **Practical implications.**
> In our work, we directly demonstrate the effectiveness of GHBM in large-scale real-world scenarios, by conducting extensive experimentation on challenging settings characterized by a large number of clients (on the order of $10^4$-$10^5$) and very low participation (e.g. $C<1\\%$).
> As the reviewer mentioned that our algorithm is simple, we want to emphasize that **the simplicity of our algorithm is a significant advantage**. Indeed, a simple, principled approach with strong theoretical guarantees and solid empirical performance, that can efficiently scale to realistic settings, is undeniably preferable to more complex alternatives.
>
> **Motivated by the above points, we believe the significance of our work is well supported.** We are confident that our work addresses a significant gap in the existing literature and offers practical benefits for FL applications.  We hope this explanation resolves the reviewer’s concerns and are happy to provide further clarifications if needed.

---

> ### Author Response · Authors · 2025-04-03
> **Author Response 2/3**
>
> **W2: Comparison with FedACG**
> We provide a comparison with the FedACG algorithm proposed in [1] based on: algorithmic design, theoretical guarantees, and empirical results.
>
> Algorithmically, it has two modifications w.r.t. FedAvgM: (i) it uses the Nesterov Accelerated Gradient (NAG) to broadcast a lookahead global model and (ii) adds a proximal local penalty similar to FedProx w.r.t. this transmitted global model. The method has the same communication complexity as FedAvg because it does not exchange additional information.
> Our work proposes instead a novel formulation of momentum, explicitly designed to provide an advantage in heterogeneous FL with partial client participation. We propose both the main algorithm (GHBM), which has _stateless_ clients but has $1.5\times$ the communication complexity of FedAvg, and communication efficient versions (e.g. FedHBM), that preserve the communication complexity as FedAvg, at the cost of using local storage.
>
> From a theoretical perspective, the convergence rate of FedACG does not prove any advantage w.r.t heterogeneity, since it still relies on the bounded heterogeneity assumption. GHBM is proven to converge under arbitrary heterogeneity in cyclic partial participation, recovering the same convergence rate that [2] proved for FedCM when in full participation. This is a significant advantage that then reflects in significantly improved performance.
>
> From an empirical perspective, we added the simulation results for FedACG in Table 2 and Fig. 7. While it is faster than FedAvgM, it still falls short of our algorithms, particularly in heterogeneous scenarios. This is a consequence of the same issue we showed in section 3.3 for classical momentum, which prevents the momentum from being updated with the contribution of all clients.
>
> **Action on the manuscript:** we added FedACG to the related works and the above full discussion in section A.1. We added results in Table 2 and convergence curves in Fig 7 in the appendix, as requested by the reviewer.
>
> [1] Communication-Efficient Federated Learning with Accelerated Client Gradient. CVPR 2024\
> [2] Cheng et al., Momentum Benefits Non-IID Federated Learning Simply and Provably, ICLR 2024

---

> ### Author Response · Authors · 2025-04-03
> **Author Response 3/3**
>
> **W3: On the claim about GHBM communication efficiency.**
> The claim that FedHBM is communication-efficient refers to the _total_ number of bytes sent over the network to reach the final model quality of FedAvg, not just to the number of bytes sent for each individual communication round. Please note that **this metric accurately reflects communication efficiency, since it takes into account both the communication complexity of the methods** (i.e. how many bytes are sent _in a single round_) **and the convergence speed** (further details on calculation are provided in section C.3).
> FedHBM displays an impressive reduction in communication cost of $87.4\\%$ with CNN and $64.1\\%$ with ResNet-20 on Cifar-100, and $67\\%$ and $51.5\\%$ on GLDv2 respectively with MobileNetV2 and ViT-B\16, which are much higher than competitor methods. In particular:
> - FedAvgM only leads to a reduction of $6.5\\%$ with CNN and $10.7\\%$ with ResNet-20 on Cifar-100, and $18\\%$ and $16.7\\%$ on GLDv2 respectively with MobileNetV2 and ViT-B\16. This is very far from the above results for FedHBM, so **FedHBM does actually provide significant communication savings w.r.t. FedAvgM**. In fact, considering only the amount of data set at each round is insufficient to draw conclusions, because that does not take into account the effectiveness of the algorithm in optimization.
> - FedHBM requires stateful clients, but it is not the only solution we propose. In particular, in the paper, we provide guidelines to choose the best-suited algorithm for the task at hand between FedHBM and GHBM (see section 3.4). For any case in which it is undesirable to have stateful clients, GHBM is the algorithm of choice. Despite communicating more bytes per round, GHBM is still more communication efficient than competitors because it is much faster and robust to heterogeneity, so the total number of bytes exchanged to reach the target accuracy remains much lower (as shown in Table 4).
>
> **Action on the manuscript:** As requested by the reviewer, we made it clearer that FedHBM has the same communication complexity as FedAvgM (server momentum) - meaning that they exchange the same amount of data _at each single round_. The advantages of FedHBM in communication efficiency follow from the much higher convergence speed w.r.t. to other methods, which greatly reduces the total number of bytes sent over the network to reach a target accuracy. We also added pointers in the paper to the second paragraph of section 3.4, which already discussed the implications of requiring stateful clients in FL for FedHBM.

---

### Author Response · Authors · 2025-04-04
**Revision uploaded**

We sincerely thank the reviewers for their valuable feedback and suggestions. We have carefully addressed each of the individual concerns in our responses to the reviews. Additionally, we have uploaded a revised version of the manuscript incorporating all requested changes. For clarity, the added or modified sections are highlighted in violet.

The Authors

---

### Decision · Action_Editor_mBck · 2025-05-29

**Recommendation:** Accept as is

**Comment:**

All reviewers give positive final recommendations (two leaning accept and one accept). After author response and revision, all major concerns raised by the reviewers have been properly addressed, including the metric of communication (as well as arguments around reduced communication), comparing with additional existing works, heterogeneity simulation, and clarification questions. Therefore, I recommend accept as is (after fixing some minor presentation issues throughout the paper, examples listed below).

Some example possible changes:
* assumption x -> Assumption x
* eq. (x) -> Eq. (x)
* make the uppercase/lowercase of the first characters of section/paragraph headers consistent
* Kim et al. (2024) uses -> Kim et al. (2024) use

**Audience:**

Yes, this paper is of interest to authors in areas such as optimization and distributed learning.

**Claims And Evidence:**

The claims (both theoretical and empirical) are supported by accurate evidence.